



# Size, shape and orientation matter: fast and automatic measurement of grain geometries from 3D point clouds

Philippe Steer[1,*], Laure Guerit[1,*], Dimitri Lague[1], Alain Crave[1], Aurélie Gourdon[1]

[1]Univ Rennes, CNRS, Géosciences Rennes - UMR 6118, 35000, Rennes, France.
*These authors contributed equally to this work.

*Correspondence to*: Philippe Steer (philippe.steer@univ-rennes1.fr) and Laure Guerit (laure.guerit@univ-rennes1.fr)

**Abstract.** The grain-scale morphology of sediments and their size distribution inform on their transport history, are important factors controlling the efficiency of erosion and transport and control the quality of aquatic ecosystems. In turn, constraining the spatial evolution of the size and shape of grains can offer deep insights on the dynamics of erosion and sediment transport in coastal, hillslope and fluvial environments. However, the size distribution of sediments is generally assessed using insufficiently representative field measurements and determining the grain-scale shape of sediments remains a real challenge in geomorphology. Here we determine the size distribution and grain-scale shape of sediments located in coastal and river environments with a new methodological approach based on the segmentation and geomorphological fitting of 3D point clouds. Point cloud segmentation into individual grains is performed using a watershed algorithm applied here to 3D point clouds. Once the grains are individualized into several sub-clouds, each grain-scale morphology is determined by fitting a 3D geometrical model applied to each sub-cloud. If different geometrical models can be conceived and tested, including cuboids and ellipsoids, this study focuses mostly on ellipsoids. A phase of results checking is then performed to remove grains showing a best-fitting model with a low level of confidence. The main benefits of this automatic and non-destructive method are that it provides access to 1) an un-biased estimate of surface grain-size distribution on a large range of scales, from centimeters to meters; 2) a very large number of data, only limited by the number of grains in the point-cloud dataset; 3) the 3D morphology of grains, in turn allowing to develop new metrics characterizing the size and shape of grains; and 4) the in-situ orientation and organization of grains and grain clusters. The main limit of this method is that it is only able to detect grains with a characteristic size greater than the resolution of the point cloud.

## 1 Introduction

Rock particles or grains are characterized by a large range of size, from clays to large boulders, and a large variety of shape and angularity, from spherical or ellipsoidal to cubic or polyhedral (e.g., Blott and Pye, 2008; Domokos et al., 2014; Domokos et al., 2020). Grains are initially formed by fragmentation or chemical weathering, transforming a cohesive rock mass into a granular material. The size and shape of grains then evolve due to the action of geomorphological processes, including attrition, chipping, abrasion, fragmentation and chemical weathering, during transport by wind, river flow, avalanches along hillslopes



or sea waves and currents (e.g., Attal and Lavé, 2006; 2009; Domokos et al., 2014; Miller et al., 2014; Várkonyi et al., 2016; Novák-Szabó et al., 2018; Marc et al., 2021). The size and shape distribution of grains in various natural environments can therefore be represented as an initial size or shape distribution, informing on fragmentation, weathering processes and on the structure of the rock mass (e.g., fracture density and orientation, mineral size) (e.g., Molnar et al., 2007; Garzanti et al., 2008; 5  Sklar et al., 2017; DiBiase et al., 2018; Neely and DiBiase, 2020; Verdian et al., 2021). These initial distributions are then progressively modified during transport, informing in turn on the transport processes (e.g. saltation or suspension), conditions (e.g. dense flows) and duration or length. Grains are also found at the surface of other planetary bodies or asteroids (Burke et al., 2021) and offer unique constraints on their surface conditions. A striking example is the use of the shape of grains to reconstruct the transport history of pebbles on Mars (Szabo et al., 2015). Moreover, the in-situ orientation of grains found in 10  deposits can also inform on the paleo-flow conditions during sediment deposition (e.g., Johansson, 1963; Rust, 1972).

The distribution of grain size, shape and orientations strongly control the dynamics of fluvial and sedimentary environments. At the scale of rivers, the size of the sediments strongly controls the mobility of alluvial grains and their incipient threshold of motion (e.g., Shields, 1936), the timescale required to mobilize landslide-driven sediments (e.g., Croissant et al., 2017), the rate of river bedrock incision through the tool-and-cover effect (Sklar and Dietrich, 2004), the width of river channels (e.g., 15  Finnegan et al., 2007; Baynes et al., 2020), or the rate of knickpoint propagation (Cook et al., 2013). At the scale of a sedimentary basin, the size of grains influences the stratigraphy of the basin together with the chemical and mechanical properties of the sediment (e.g., Armitage et al., 2011). Grain size, shape and orientation in riverbeds are also key factors for aquatic habitats (e.g., Kondolf and Wolman, 1993; Riebe et al., 2014), for water and nutrient exchange through the hyporheic zone (e.g., Tonina and Buffington, 2009) or even for river hydraulics by impacting basal friction (e.g., Hodge et al., 2009).

20  Despite the ubiquitous role of grain geometry on landscape properties and dynamics, and its potentiality to constrain paleo-conditions on Earth and other planetary bodies, the 3D geometry of grains and their statistical distributions in natural environments remain poorly known. Sampling the grain-size distribution of the sediments lying at the surface of a riverbed is most often done by the grid-by-number method (Wolman, 1954). This method consists in measuring the diameter of a pre-defined number of grains, generally greater than 100. The grid-by-number method, which is simple to implement, is considered 25  as directly similar to a volumetric sampling (see Bunte and Abt, 2001; and references therein). It is therefore still widely used on the field (e.g., D'Arcy et al., 2017; Guerit et al., 2014; 2018; Chen et al., 2018; Roda-Bolua et al., 2018; Watkins et al., 2020; Baynes et al., 2020). However, samples are often taken over a few squared meters and thus lead to inherent representativity bias and to statistical bias, associated to the operator, the grain sampling strategy, the measurements themselves and to the choice of the diameter to be measured. Collection of data set can be extremely time consuming, especially when 30  many grains have to be measured to be statistically significant (Rice and Church, 1996; Green, 2003; Eaton et al., 2019). Measurements are also partly destructive (i.e., grains are moved), which generally lead to information being lost on grain orientation and exact location.

These issues have led to the development of alternative methods based on image analysis to characterize large areas in a manageable amount of time. Object-based and statistical-based approaches have been developed to characterize grain-size



distributions from pictures or 3D data. The first one (so-called "picture-sieving") consists in measuring each grain or a number of selected grains on a picture (e.g., Bunte and Abt, 2001). Several algorithms now exist to perform these measurements manually, directly on a picture (Roduit, 2008). Because this procedure can be quite time consuming, semi-automatic to automatic procedures have been implemented to automatically recognize grains from pictures (Graham et al., 2005a,b; Detert

and Weitbrecht, 2012; Buscombe et al., 2013; Purinton and Bookhagen, 2019, Soloy et al., 2020). The second approach is based on image-texture analyses and aims at correlating some statistical properties of images with the median grain size of the study site (Buscombe and Masselink, 2009; Buscombe et al., 2010; Rubin, 2004; Carbonneau et al., 2004). Similarly, 3D approaches relating empirically bed roughness, measured on high-resolution topographic data, can be implemented to infer the grain-size distribution from locally calibrated relationships (e.g., Rychkov et al., 2012; Westoby et al., 2015; Woodget and

Austrums, 2017, Vazquez-Tarrio et al., 2017; Pearson et al., 2017; Groom et al., 2018; Detert et al., 2018). These approaches considerably reduce the time spend on the field, increase efficiently the sampling density and coverage, and are non-destructive. Yet, post-processing remains time-consuming, and these methods are inherently limited to the 2D measurement of apparent axis (Graham et al., 2010) of individual grains, or to empirical local correlations with little generalization capability and limited potential to fully explore the 3D geometry of individual grains.

The last decade has seen a steep growth in the use of high-resolution 3D topographic data in Earth Sciences and geomorphology, obtained by LiDAR measurements and photogrammetry (e.g., Schneider et al., 2015; Westoby et al., 2012; Leduc et al., 2019). The resulting 3D point clouds offer unprecedented access to landscape heterogeneities and to landscape temporal evolution (e.g., Hodge et al., 2009; Leyland et al., 2017; Beer et al., 2017; Bernard et al., 2021). The accessibility of 3D point clouds, obtained from terrestrial, drone and airborne data, and their ability to capture object geometries robustly and

accurately in 3D at various scales represent a timely opportunity to develop point cloud-based methods to the issue of grain size measurement. In this paper, we develop an automatic and efficient method, entitled G3Point (standing for "Granulometry from 3D Point clouds"), to measure grain size, shape and orientation using 3D point clouds. The associated workflow consists in the 3D segmentation of individual grains using a type of watershed algorithm, the geometrical description of individual grains using 3D ellipsoidal models, and the description of the 3D geometry of the grain population using statistical

distributions. After describing the new method, we test it against synthetic and natural controlled experiments (e.g., riverbeds and beaches), considering point clouds obtained with Structure From Motion (SFM) to check its ability to robustly capture the 3D geometry and size of grains.

## 2 Method

G3Point is a Matlab program which aims at measuring the size, shape, and orientation of a large number of individual grains

as detected from any type of 3D point clouds describing the topography of surfaces covered by sediments. The main functions of G3Points are described in the following and summarized by Figure 1. A 3D point cloud represents a topographic surface defined by a set of points associated to a 3D coordinate system. Compared to 2D digital elevation models where elevation $z$ is





defined as a function of 2 horizontal coordinates $(x, y)$, 3D point clouds can include several points located at the same horizontal position (e.g., the face above and below a grain), allowing a better description of geomorphological features such as grains. In the following, we will assume that the considered point cloud is already denoised and classified to remove points associated to vegetation or other features unrelated to the sediment cover. Several efficient algorithms are available to perform

this task (e.g., Lague and Brodu, 2013). We also assume that the point cloud surface, over the region of interest (i.e., generally an area of a few 10 $m^2$, what we later refer to as the "patch-scale"), is relatively planar with its normal orientated vertically upward. We provide functions to denoise and re-orientate the point cloud accordingly. To illustrate the method, we will apply it to a point cloud of an active alluvial riverbed, of area ~40 $m^2$, acquired in 2011 with a terrestrial LiDAR scanner (Leica ScanStation 2) along the Otira River in New Zealand (Fig. 2) and already featured in Brodu and Lague (2012). The subset of

this point cloud that we use in the following is made of ~$10^5$ points for an average resolution of ~2.4 $10^3$ point/$m^2$ and was obtained after a single scan (Fig. 2a).

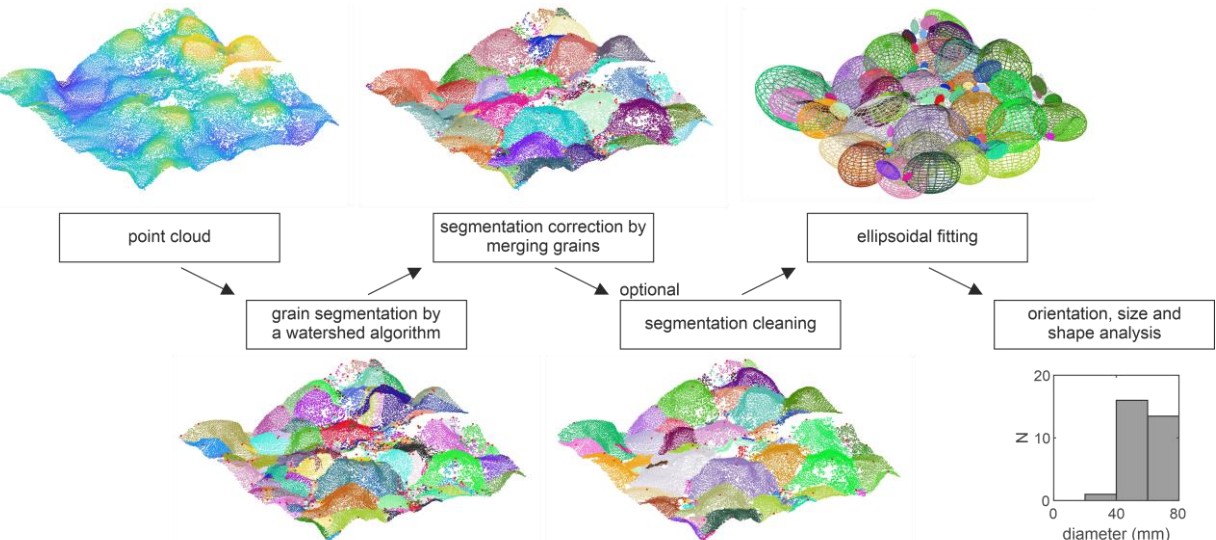

**Figure 1.** Overview of the G3Point algorithm showing the main series of functions (center) and the results (top and bottom figures). Each main function is described in detail in the Method section.

**2.1 Initial segmentation: from a global point cloud to individual grains using a watershed algorithm**

The segmentation of the point clouds into sub-point clouds representing individual grains is performed using a single flow algorithm based on the steepest slope criterion (O'Callaghan and Mark, 1984). This algorithm is generally used to route water and identify watersheds on 2D Digital Elevation Models (DEM). It uses the steepest slope criterion to route water between neighborhood points until reaching a local minimum, which corresponds to the outlet of the watershed. Each watershed is

therefore described by a directed acyclic graph which associates each point of the point cloud to its outlet node through a single flow path (e.g., Schwanghart and Scherler, 2014). The Fastscape algorithm offers a fast solution to order points along the steepest water flow path (Braun and Willett, 2013). This algorithm can be readily adapted to irregular grids, such as 3D point



clouds, as long as the neighborhood nodes of each node is known. We use here the $k$-nearest neighbors algorithm, using 3D distances, to identify the neighborhood nodes. The parameter $k$ controls the "neighborhood scale" which varies locally based on the spatial density of points. For the point cloud of the Otira River, $k$ was taken equal to 20.

To identify grains instead of watersheds, the single flow algorithm is modified by using the criterion of the steepest slope upward instead of the steepest slope downward to route water. In other words, water is routed from a point to its steepest upward neighbor, which is associated to the maximum value of $\Delta z/(\Delta x^2 + \Delta y^2)^{1/2}$, with $\Delta x$, $\Delta y$ and $\Delta z$ the distance along the $x$, $y$ and $z$ between the considered point and its $k$-nearest neighbors. Using this approach, each grain is theoretically identified by a single watershed, and the associated outlet corresponds to the summit of the grain. For the Otira River, the initial segmentation identifies 772 grains (Fig. 2b), and their set of points are associated to a unique label. This segmentation approach is convenient as it is fast (i.e. ~0.1s or ~1s of CPU time on a laptop for ~$10^5$ or ~$10^6$ points, respectively), relatively simple to implement, and the topology of a grain can be simplified to the position of its summit (red dots on Fig. 2b). Moreover, this algorithm only imposes one scale: the theoretical minimum grain diameter which can be segmented, i.e., the local neighborhood scale. Except for the neighborhood scale, no other scale is introduced, and the algorithm can identify grains of varying size. However, results show that this watershed segmentation approach also leads to a global over-segmentation of grains. Indeed, grains can exhibit several local maxima, due to the geometry of the grain or to a rough surface or to potential data noise, leading to a grain being over-segmented (Fig. 2b).

## 2.2 Correcting from over-segmentation by merging grains

Correcting over-segmentation is not a trivial task due the large range of grain sizes and classical clustering approach, as hierarchical clustering or dbscan (density-based spatial clustering of applications with noise) (e.g., Esther et al., 1996) proved ineffective to solve for this issue. Moreover, using approaches that use all the points of the point cloud can lead to significant computational time which might become prohibitive for large point clouds. Here, we develop an approach which makes use of the properties of the segmented watersheds, which associate grains (i.e., watersheds) to their unique summit points (i.e., outlets) and to their border nodes (i.e., crests). We combine 3 criteria to decide if a pair of grains $(i, j)$ should be merged in a single grain.

1. Criteria 1: The distance $d_{ij}$ between two summit points should be smaller than the sum of the characteristic radius of the two grains. Instead of using a criterion based on a single scale to decide whether two grains should be merged, which would be problematic due to the large range of grain size, we use the drainage area $A$ at the summit node (i.e., outlet), which receives water from all the points sharing the same label, to determine a characteristic scale or grain radius $l_i = (A_i/\pi)^{1/2}$. The criterion to merge the pair of grains $(i, j)$ together is therefore $d_{ij} < C_F(l_i + l_j)$, with $C_F$ a factor that we take generally equal to 0.5-1. These values were obtained after several trial-and-error tests.

2. Criteria 2: Grains $i$ and $j$ should be neighbors (i.e., at least one of the points of grain $i$ belongs to the neighborhood of the points of the grain $j$, and vice versa)



3.  Criteria 3: The 3D angle between the normals of the crest points of grains $i$ and $j$ should be small. Orientation of the normal is computed by taking the normal of the best fitting local plane to the $k$-nearest neighbors of the considered point. For each of the crest node of grain $i$, the sum of the 3D angle between its normal and the normal of its neighbors belonging to grain $j$ is computed. This operation is performed for every crest point of grains $i$ and $j$, and then a mean 3D angle is determined. The criterion to merge the grains is that their mean 3D angle is lower than a threshold $\alpha$ that we take equals to 60˚ in the following. This last criterion prevents grains that are clearly separated by a curved border to be merged.

Therefore, a pair of grains $(i, j)$ is merged if and only if these three criteria are respected. Due to the low number of grains, compared to the number of points in the point cloud, this step is also fast (i.e. ~0.1-1s or ~1-10s of CPU time on a laptop for ~$10^5$ or ~$10^6$ points, respectively). The results show that many labels, suffering from over-segmentation and describing a single grain, were effectively merged by applying this test, leaving only 657 labels or grains instead of 772 (Fig. 2c). Overall, the resulting segmentation looks qualitatively good, even if some grains still suffer from over-segmentation while a limited number of labels now suffer from under-segmentation and include more than one grain.

## 2.3 Segmentation cleaning operations

If this initial segmentation is deemed satisfactory at first order, some minor flaws can lead to an inaccurate description of the geometry of grains and their size distributions. To increase the quality of the segmentation, we optionally offer routines to perform several post-segmentation operations:

1)  Applying Criteria 3 only, which consists in merging a pair of grains if the 3D angle between their normal, computed on the common border, is lower than a threshold $\beta$. The objective is mostly to merge small grains, resulting from the initial over-segmentation due to grain local maxima, with large ones.

2)  Cleaning the segmentation by removing grains with less than $n_{min}$ points. This number of points should be greater or equal greater than the number of nearest neighbors $k$ and 10, considered as the strict minimum number of points required to fit an ellipsoid (i.e., number of parameters of an ellipsoid). However, larger values of $n_{min}$ should be favoured to reduce the uncertainty of the resulting ellipsoidal model.

3)  Removing flattish or over-elongated grains as they generally do not correspond to individual grains but to clusters of fine grains with a characteristic size much lower than the typical point spacing of classical point clouds or to unproperly segmented grains, respectively. To detect flattish or over-elongated grains, we perform a singular value decomposition (SVD) over the 3D coordinates of each of the sub-point clouds. If a grain has a minimum or an intermediate singular value divided by its maximum singular value lower than a threshold $\emptyset_{flat}$ or $2\emptyset_{flat}$, then this grain is considered flattish or over-elongated, respectively, and removed from the segmentation. Values of $\emptyset_{flat} <$ 0.1 were found to be suitable in this study, even if natural settings with very flat (e.g., as found for slate grains) or elongated grains should probably consider smaller values.

In the example shown in figure 2, the segmentation was not cleaned.



**Figure 2.** 3D view of the point cloud, its segmentation into individual grains and the fitted ellipsoids. a) Initial point cloud with the colormap indicating the elevation of the points. b) Initial segmentation of the point cloud into individual grains performed with a modified watershed algorithm using the steepest slope upward criterion to route water. c) Segmentation after merging close grains together. d) Ellipsoids fitted





to each individual grains identified on panel c are represented with colored lines (same color than for panel c) over the point cloud (black dots). Color in panels a, b and c indicates the label of the grains (i.e., one color per grain). Red dots on panels a and b indicate the location of the summit point of each grain. e) Picture showing the location of the point cloud surface, bounded by a red polygon, relatively to the Otira river.

**2.4 Geometrical modelling: 3D ellipsoidal fitting of grains**

Once the grains are segmented and labelled, the following phase consists in the 3D geometrical description of the geometry of each of the grains. We particularly seek to extract their 3D size and orientation, and to infer an overall adequacy to simple shapes. A strong constraint results from the fact that only an unknown fraction of the upper surface of the segmented grains (i.e., the visible part of the grain) is topographically described by the point cloud. This prevents us to directly use the point cloud describing each grain to measure their size and orientation. Instead, we rely on the use of geometrical models to represent each grain. The most pertinent and simplest 3D geometrical model to describe a grain is the ellipsoidal model. Two strategies are adopted to describe the geometry of a grain with an ellipsoidal model: fitting an ellipsoid or determining its ellipsoid of inertia.

Fitting an ellipsoid to a set of points in 3D is a complex problem that has received attention from different applied mathematics communities, including computer vision, pattern recognition, numerical analysis, and statistics. Ellipsoids belong to the family of quadric surfaces that can be defined as:

$$Ax^2 + By^2 + Cz^2 + 2Fyz + 2Gxz + 2Hxy + 2Px + 2Qy + 2Rz + D = 0, (1)$$

where $A$, $B$, $C$, $F$, G, $H$, P, Q, R and $D$ are the parameters of the quadric surface. Defining $I = A + B + C$ and $J = AB + BC + AC - F^2 - G^2 - H^2$, it can be shown that equation (1) must represent an ellipsoid when $4J - I^2 > 0$ (Li and Griffiths, 2004). This condition is respected when the short radius is at least half the length of the major radius of an ellipsoid. This represents a sufficient condition, but not a necessary one, and ellipsoids can be mathematically defined without respecting $4J - I^2 > 0$. Anyhow, we use an efficient and robust Matlab version (Hunyadi, 2022) of a direct least square fitting method (Li and Griffiths, 2004), based on the condition that $4J - I^2 > 0$, to describe the geometry of the segmented grains by minimizing the square of the distance between labeled points and the ellipsoidal model. For ellipsoids fitting grains which do not respect this condition, the fitting method might still lead to ellipsoids or to other quadric surfaces. Grains suffering from fitting issues or leading to quadric surfaces other than an ellipsoid are filtered out, leaving 630 correctly fitted ellipsoids over 657 labelled grains. The resulting ellipsoids, fitted to each labelled grain, appear qualitatively consistent with the shape, size and orientation of the labelled grains (Fig. 2d). Other ellipsoidal fitting algorithms exist, but this direct least-square approach was found to lead to the best solution. In turn, the condition $4J - I^2 > 0$ prevents the occurrence of flat or over-elongated ellipsoids, which could otherwise represent better mathematical solutions despite being, in some cases, physically unlikely.

The second approach considered to characterize the geometry of the grains consists in computing the inertia ellipsoids corresponding to the labelled points of the grains. This is performed, first by computing the mean position of the points, second by computing the covariance matrix of the points subtracted from their mean position, and third by making a singular value decomposition of the covariance matrix normalized by the number of points.




The approach based on the inertia ellipsoid can be considered simpler than the direct least-square fitting method and does not suffer from mathematical constraints of the direct least-square approach. However, as it is not a fitting method, its main drawback is that it is unable to guess the "hidden" geometry of the grains (i.e., by using the curvature of the visible part of the grain), and the obtained inertia ellipsoids will tend to be flatter than the grains. We later compare the two approaches in the Results section. We also compare the obtained ellipsoids to cuboids that are obtained by determining the minimal 3D bounding box for each grain, with at least one side oriented along the horizontal plan.

## 2.5 Geometrical and statistical description of grain size, shape and orientation

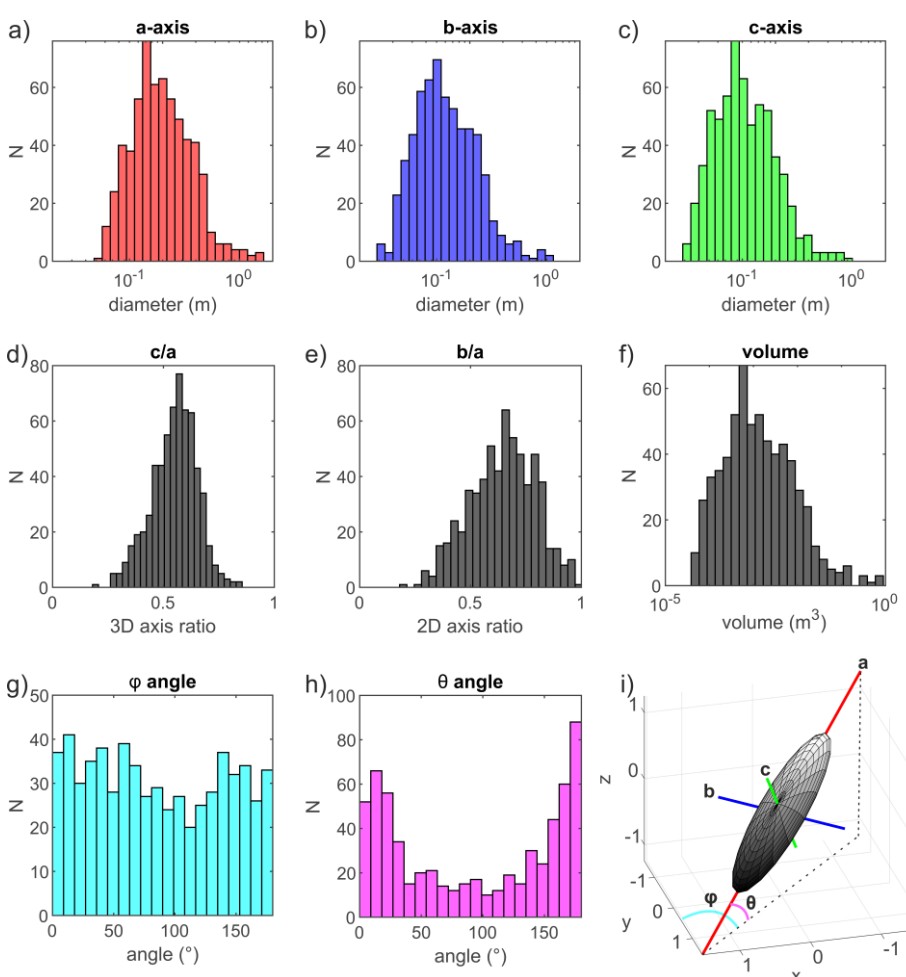

**Figure 3.** Size, shape and orientation distribution of 630 ellipsoids correctly fitted to the labelled grains. Histogram distribution of the diameters of the ellipsoids along their a) major $a-$, b) intermediate $b-$ and c) short $c-$ axis. Histogram distribution of the d) 3D axis ratio ($c/a$), e) 2D axis ratio ($b/a$) and f) volume of the ellipsoids. Histogram distribution of the g) azimuth $\varphi$ and h) dip $\theta$ angle. i) 3D view of an arbitrary ellipsoid and representation of the different metrics used to characterize ellipsoid size, shape and orientation.



Once the grains are fitted by an ellipsoid, it is straightforward to access their geometrical information. For each ellipsoid, we measure the radius (and the diameter, as classically used for grain-size distributions) of the major $a-$, intermediate $b-$ and short $c-$axis, the orientation (i.e., azimut and dip) of these 3 axis, the volume of the ellipsoid $V = 4/3\,\pi abc$ and the approximate surface area $S$ of the ellipsoid using Knud Thomsen's formula $S = 4\pi((a^p b^p + a^p c^p + b^p c^p)/3)^{1/p}$. Indeed,

there is no general formula for estimating $S$ and this formula approximate the ellipsoid area with an error less than 1.061 % when $p = 1.6705$. We can also compute 2 different axis ratios, with $c/a$ the 3D axis ratio between the short and major axis, and $b/a$ the 2D axis ratio (or elongation ratio) between the intermediate and the major axis. We coin this latter the 2D axis ratio as it generally corresponds to the axis ratio measured from 2D images, by contrast with the 3D axis ratio that is generally not measurable from 2D images (i.e., assuming that the short axis is oriented vertically). Other metrics can be computed such

as the grain intercept sphericity defined as $\psi = \left(\frac{bc}{a^2}\right)^{1/3}$ (Krumbein, 1941; Bunte and Abt, 2001), which varies between 0 (i.e., non-spherical) and 1 (i.e., spherical). In the following, we will refer to this metrics as being the sphericity.

For each grain, we can also compute the distance of each point of the grain, of coordinates $(x, y, z)$, to its projection on the ellipsoid surface, of coordinates $(x_e, y_e, z_e)$. The square of this distance, corresponding to the residuals in a least-square sense, characterizes the goodness of the fit through the coefficient of determination: $R^2 = 1 -$

$\sum((x - x_e)^2 + (y - y_e)^2 + (z - z_e)^2)/\sum\left(\left(x - \bar{x}\right)^2 + \left(y - \bar{y}\right)^2 + \left(z - \bar{z}\right)^2\right)$, with $\bar{x}$, $\bar{y}$ and $\bar{z}$ the mean coordinates of the points. $R^2$ informs on the quality of the mathematical fit itself and on the consistency between the ellipsoidal model and the shape of the grain, which can deviate significantly from an ellipsoidal geometry.

The statistical description of grain geometrical properties of a grain population, such as the classical 1D grain-size distribution (GSD), is then performed based on the geometrical attributes of each individual grain of the considered population (Fig. 3).

The range of measured diameter, ~0.01 to ~1 m, span two orders of magnitude (Fig. 3a-c), and the 3D (c/a) and 2D (b/a) axis ratios unsurprisingly vary between 0 and 1 with mean values of 0.55 and 0.65, respectively (Fig. 3d-e). The range of volume of the ellipsoids spans almost 5 orders of magnitude, from $10^{-5}$ to 1 m$^3$ (Fig. 3f). In addition to this classic description, G3Point also provides information on the 3D organization of the grains. Here, the orientation distribution of the grains along this active alluvial bed shows that there is no preferential orientation of grains due to the river flow, as they appear to follow a mostly

uniform distribution of the azimuth $\varphi$ (Fig. 3g) and that most grains are lying, as testified by their dip angle $\theta$, in a sub-horizontal position with $0 < \theta < 30°$ or $150 < \theta < 180°$ (Fig. 3h).

## 3 Results: method validation and application to synthetic or natural environments

In addition of its robustness and efficiency, an algorithm dedicated to extract granulometric information from point clouds must be able to manage various sources of data, including SFM and LiDAR. In the following, we therefore test the newly

developed algorithm against "ground truth" datasets of grain size, obtained in synthetic or natural environments. For each data set, we compare the distribution obtained with G3Point to the grain-size distribution measured by hand.





## 3.1 Synthetic environment: the test of the pebbles on a flat surface



**Figure 4.** Results from the synthetic experiment considering 39 pebbles on a flat surface. a) Point cloud (grey dots) of the experiment overlayed by the label (color) of each identified grain and their summit node (red dot). The resulting b) cuboid (red) and ellipsoids obtained with c) a direct least square (DLSF, blue) and d) the inertia ellipsoid (IR, green) approaches. Diameters measured along the e) $a - axis$ (top), $b - axis$ (middle) and $c -$axis (bottom) using the direct least square (DLSF, blue dots) and the inertia ellipsoid (IE, green dots) approaches for the 39 grains as a function of the cuboid lengths. The red dots show the dimensions of the average ellipsoid between the IE and DSLF ellipsoids. f) Axis ratios of the ellipsoids as a function of the axis ratios of the cuboids. g) Volume, area and azimuthal angle of the $a -$axis (0-180˚) of the ellipsoids as a function of the azimuthal angle of the cuboids. The black dashed lines show the 1:1 line on all the panels.

The first experiment consists in 39 black pebbles, bought in a hardware store, laying in a horizontal position over a planar surface of 0.5 x 0.5 m (Fig. 4a). This synthetic experiment was captured by pictures to generate a 3D point cloud by SFM. Data were processed with Agisoft Metashape and the resulting point cloud, made of ~2 $10^5$ points, has a native resolution of ~1 point per millimeter. To segment grains, and only grains, the planar surface is removed from the point cloud. G3Point is then applied to this point cloud using the couple of parameters $k = 100$ and $C_F = 0.8$, which was found satisfying after a trial-



and-error series of tests. Indeed, the 39 pebbles are perfectly detected and labelled as individual grains. Each grain is then described by a cuboid (Fig. 4b) and ellipsoidal models using the direct least square fitting method (DLSF) (Fig. 4c), as previously done, and the inertia ellipsoid (IE) approach (Fig. 4d). We force the vertical dimension of the cuboids to start, for their lower face, at the elevation of the planar surface, to correctly capture the height of the grains. The major $a-$, intermediate

$b-$ and short $c-$ axes of the modelled ellipsoids are then compared to the true diameters of the pebbles, which are assumed to be characterized by the length, width and height of the cuboids, respectively. We emphasize here that most of the pebbles used for this test are strongly elongated ($b/a{\sim}0.5$) and flat ($c/a{\sim}0.25$), which can represent real challenges for most ellipsoidal fitting algorithms. This test should therefore be considered as an end-member scenario, testing the ability of the approach to properly describe the geometry of grains using ellipsoidal models.

Despite that, the obtained diameters for the $a-$, $b-$ and $c-$axes are roughly consistent in between the 3 approaches (Fig. 4e), even if the diameters obtained with the DLSF and IE approaches are almost systematically higher or lower, respectively, than the cuboid dimensions. The ratios between the ellipsoid diameters and the cuboid lengths for the $a-$ and $b-$axis range between 0.8 and 1 for the IE and between 0.8 and 2 for the DLSF (see Fig. S1 in the Supplement). For the $c-$axis, the consistency is less good and the ratio range between 0.4-0.9 and 1.1-9 for the IE and DLSF approaches, respectively. The

results reflect the pros and cons of each approach: the DLSF approach leads to larger than expected ellipsoids, due to the geometrical constrain of the fitting algorithm for the $c-$axis, while the IE approach leads to smaller than expected ellipsoids, as only the upper face of the grains is accounted for. This is well illustrated by the difference in the resulting 3D ($c/a$) and 2D ($b/a$) axis ratio. If the 2D axis ratio is relatively consistent in between the three approaches (Fig. 4f), the 3D axis ratio of the DLSF ellipsoids (0.4-0.65) is significantly higher than the one of the cuboids (0.1-0.4), except for one grain. On the contrary,

the 3D axis ratio of the IE ellipsoids is always lower than the one of the cuboids. These discrepancies also lead to a larger or lower volume and area for the DLSF or IE ellipsoids, respectively, compared to the cuboid volume and area (Fig. 4g). We note that the consistency of the DLSF ellipsoids with the cuboids is greatly improved when increasing the 3D axis ratio (i.e., when considering more spherical grains), which limits the role of the geometrical constrain on the quality of the fitted ellipsoid. Last, the horizontal orientation of the DLSF or IE ellipsoids, given by the azimuthal angle of the $a$-axis, is relatively consistent with

the orientation of the cuboids (Fig. 4g).

Despite a good first-order accuracy of the considered ellipsoidal models to represent the 3D dimensions of grains, none of these approaches is deemed systematically suitable by itself. The consistency of the ellipsoidal models with the true geometry of the grains depends on the considered geometrical model, on the surface coverage of the grain by the point cloud and on the shape of the grain itself (see Figure A1 and Appendix A). In the following, instead of relying on a single ellipsoidal model,

we rather assess the geometry and dimensions of grains by using both the DLSF and IE ellipsoidal models. Indeed, considering the size (or size distribution) obtained with the DLSF and IE ellipsoidal models offer an upper and lower bound on the true size (or size distribution) of the grain (or grain population). We also provide a mean size (or size distribution) obtained with these two ellipsoidal models to offer an approximate solution to the true size of the grain (or grain population).





## 3.2 Field experiments with SFM 3D point clouds

**Figure 5.** Field pictures (top), initial point clouds colored coded in elevation and segmented point clouds (middle) and grain-size distributions (bottom) from a) Site 1 and b) Site 2 of the Pointe du Chateau Renard and c) the Hérault River. Distributions of the a- (red), b- (green) and
5    c- (blue) axis result from Wolman counts (dark colors) and G3Point (light colors). Shaded envelops correspond to uncertainties defined by bootstrap approach for Wolman counts and by the envelop defined by the two fitting methods for G3Point (see text for details). Locations of the Wolman lines (white) and SFM covers (black polygons) are indicated on the pictures.

The second experiment consists in pebbles from three natural field sites in France, the beach of Pointe du Chateau Renard
10    (Brittany) with coarse and angular grains at Site 1 and smaller rounded grains at Site 2 (Fig. 5a-b), and the Hérault River near Saint-André-de-Majencoules (Cévennes) with rounded, fluvially-transported pebbles (Fig. 5c). At each site, we sampled the grain-size distribution by Wolman grid-by-number method (Wolman, 1954). At Site 1 of Pointe du Chateau Renard, we



defined a grid of about 2.5 x 3 m with nodes every 0.3 m, we measured the three axes of each grain lying under a node and a total of 76 grains were measured. At Site 2, we stretched two parallel decameters and wo operators walked along these lines, picked the two grains lying under each of their hands (random selection) about every meter, and measured the three axes of the grains. In total, 529 grains were measured. For the Hérault River, we defined a grid of 2.5 x 13 m with nodes every 0.4 m

5  and we measured the intermediate axis of 197 grains. Measurements were performed with a calliper and rounded toward the nearest millimeter or with a decameter and rounded toward the nearest 5 mm, for small or large grains, respectively. Only grains larger than 4 mm were measured. In addition to operator errors, related to the measurement itself and to the choice of the diameter to measure, the resulting distribution is associated with uncertainties related to the size of the sample. We used a bootstrap approach with replacement to evaluate the confidence interval of each distribution (Rice and Church, 1996; Bunte

10  and Abt, 2001; Green, 2003). For each sample, we randomly sampled 10000 replicates of the distribution and the scatter defines the confidence interval. The pebbles at Site 1 of the beach of Pointe du Chateau Renard have a median a-axis of 170+/-30 mm, a median b-axis of 110+/-20 mm and a median c-axis of 60+/-15 mm (Table 1). At Site 2, the pebbles have a median a-axis of 117+/-13 mm, a median b-axis of 80+/-8 mm and a median c-axis of 50+/-7 mm (Table 1). The fluvial pebbles along the Hérault River are smaller, with a median b-axis of 75+/-12 mm (Table 1).

| Site | Method | Number of grains | k | cf | $\alpha$ | $\beta$ | $\phi_{flat}$ | $A_{thres}$ | Min point |
|---|---|---|---|---|---|---|---|---|---|
| Chateau Renard Site 1 | Wolman | 76 | - | - | - | - | - | - | - |
| | G3Point | 80 | 30 | 0.6 | 35 | 5 | 0.2 | 10 | 50 |
| Chateau Renard Site 2 | Wolman | 529 | - | - | - | - | - | - | - |
| | G3Point | 356 | 40 | 0.5 | 40 | 10 | 0.2 | 20 | 100 |
| Hérault | Wolman | 197 | - | - | - | - | - | - | - |
| | G3Point | 192 | 50 | 0.3 | 35 | 10 | 0.1 | 20 | 100 |

**Table 1.** Statistics of the grain-size distributions for the three sites surveyed by SFM. The six coefficients (k, $C_F$, $\alpha$, $\beta$, $\phi_{flat}$, $A_{thres}$) are the parameters required for G3Point (see text for details).

At each site, we took about a hundred of pictures with a Nikon D3500 that covered a few squared meters to build a 3D point

20  cloud by SFM. Data were processed with Agisoft Metashape and the resulting point clouds have a native resolution of ~ 1 point per millimeter. We subsampled the point clouds with CloudCompare to ~ 1 point per 2 to 3 mm to reduce calculation duration. G3Point is then applied to the resulting point clouds with parameters defined after a trial-and-error series of tests so





that the segmentation of the grains is visually satisfying (Fig. 5). With this approach, a large number of grains is detected (342, 901 and 831 for Chateau Renard Site 1, Site 2 and the Hérault river, respectively, Table 1) and each segmented grain is fitted with two different ellipsoidal fits, DLSF and IE.

To compare the distributions obtained by G3Point to the distributions obtained by Wolman counts on the field, we perform synthetic Wolman samplings on the fitted grains, for each fitting approach. We apply a virtual grid to the point cloud and extract the three axes of the grains lying under the nodes, with grid spacing defined as half the maximum $b$-axis (this roughly corresponds to the D90). We now have 81, 426 and 284 grains for Chateau Renard Site 1, Site 2 and the Hérault River, respectively, close to the number of grains measured on the field at each site (Table 1). Because we can easily resample the point cloud, we repeat this operation 50 times for each fitting method and define the grain-size distribution as the average of these 50 samples. Then, the envelope defined by these two average grain-size distributions (one for DLSF, one for IE) is used as the confidence interval of each distribution, as presented in the previous subsection. We consider the average distribution obtained by these two methods as the grain-size distribution of the sample and define the median axes on this distribution (Fig. 5). The confidence intervals of the $a$- and $b$- axis are always quite narrow (i.e., within a few percents of the average value) but we observe intervals close to +/- 50 % for the $c$-axis due to the assumptions made by the fitting methods for the $c$-axis (see the Method section for details).

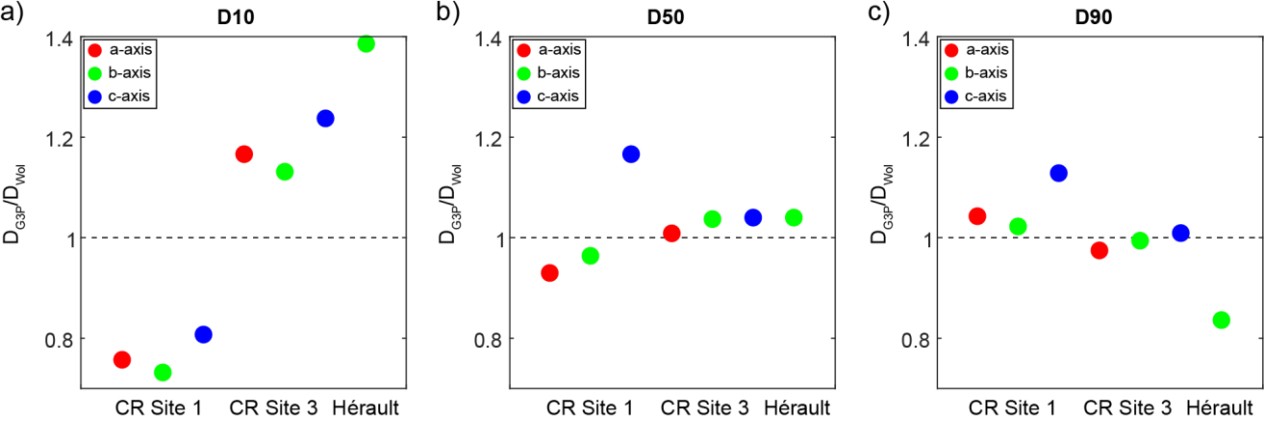

**Figure 6:** Ratio between the main quartiles (a) D10, b) D50 and c) D90) defined by Wolman counts and G3Point according to the sampling sites, for the 3 grain axes ($a$-axis: red, $b$-axis: green, $c$-axis: blue). A ratio below (above) 1 indicates an underestimation (overestimation) with G3Point with respect to field measurement.

For the three study sites, distributions obtained with G3Point are always within the uncertainties of manual counts distributions, except for the smaller quartiles. In fact, we observe that G3point systematically over- or under-estimates the 10[th] quartile (D10) of the distributions by 20 to 40 % for the three axes of the three sampling sites (Fig. 6a). We propose that this is due to the inability of the algorithm to recover small grains because their relief is too limited to be accurately segmented. However, the two methods lead to similar (i.e., always within uncertainties) median diameters for any grain axis (Table 2). In fact, based on G3point, we recover a median $a$-axis of 164+/-11 mm, a median $b$-axis of 111+/-7 mm and a median $c$-axis of 68+/-30 mm





for Pointe du Chateau Renard Site 1 (Table 2). We thus underestimate the *a*-axis D50 by 4% and we overestimate the *b*- and *c*-axis D50 by 1 %, and 13 %, respectively, with respect to field counts (Fig. 6b). This is below the uncertainties associated with field measurements in this study and below the typical uncertainties associated with manual grain-size measurements (Green, 2003). For Site 2, the median *a*-axis is 117+/-8 mm, the median *b*-axis of 82+/-9 mm and a median *c*-axis of 52+/-24 mm (Table 2). We thus recover the same *a*-axis we found with Wolman counts and we overestimate the *b*-axis D50 by 3 % and the *c*-axis D50 by 4 % with respect to field counts (Fig. 6b). For the Hérault River, we recover a median *b*-axis of 77+/-4 mm (Table 2) and thus overestimate the *b*-axis D50 by 3 % (Fig. 6b), which is again below uncertainties associated with field measurements (this study; Green, 2003). Similar accuracies are observed for the D90. In fact, at Pointe du Chateau Renard Site 1, we overestimate the D90s by 4, 1 and 8 % with G3Point with respect to Wolman counts (Fig. 6c, Table 2). At Site 2, the D90s are overestimated by 6, 2 and 2 %, and by 16 % for the Hérault River (Fig. 6c, Table 2). These numbers are always lower than the variability associated with field counts.

| Site | Method | a-axis | | | b-axis | | | c-axis | | |
|---|---|---|---|---|---|---|---|---|---|---|
| | | D10 (mm) | D50 (mm) | D90 (mm) | D10 (mm) | D50 (mm) | D90 (mm) | D10 (mm) | D50 (mm) | D90 (mm) |
| Chateau Renard Site 1 | Wolman | 82±17 | 170±51 | 304±154 | 52±17 | 110±42 | 224±91 | 31±11 | 60±19 | 132±57 |
| | G3Point | 62±4 | 158±9 | 317±46 | 38±6 | 106±10 | 229±23 | 25±13 | 70±31 | 149±53 |
| Chateau Renard Site 2 | Wolman | 54±7 | 117±15 | 235±28 | 38±6 | 81±10 | 165±22 | 21±4 | 50±8 | 110±19 |
| | G3Point | 63±3 | 118±8 | 229±19 | 43±5 | 84±10 | 164±14 | 26±14 | 52±24 | 111±41 |
| Hérault | Wolman | - | - | - | 31±13 | 75±18 | 164±44 | - | - | - |
| | G3Point | - | - | - | 43±3 | 78±5 | 137±10 | - | - | - |

**Table 2.** Characteristic quartiles of the grain-size distributions obtained at the three sites by Wolman counts and with G3Point. D10, D50 and D90 are the 10[th], 50[th] and 90[th] quartiles of the distribution, respectively. The *a*-, *b*- and *c*- axis are the large, intermediate and small axis of the grains, respectively.

This second experiment based on natural grains thus confirms that G3Point is efficient at recovering Wolman-like grain-size distributions for pebble and cobble populations in different environments and for various grain angularity, with a limited temporal cost on the field and in the lab. The best performance of the algorithm is for the median and coarse quartiles (D50 and above).



## 4 Discussions

### 4.1 Practical considerations for using G3Point

As already demonstrated, G3Point is designed to perform automatic 3D granulometric measurements on point clouds over surface area 1-100 m$^2$ (hereinafter referred to the "patch-scale") with a typical resolution of ~0.1-1 cm/point and a total number of points around 10$^6$. This scale enables 1) to perform efficient and fast measurements (i.e., several seconds), 2) to visually check the quality of the resulting segmentation of the grains and 3) to compare the resulting grain-size distribution with the one obtained with manual counting. We therefore suggest using G3Point mostly for patch-scale studies. However, G3Point can also perform grain size, shape and orientation analysis over larger study area (> 100 m$^2$). In this case, the best practice consists in segmenting the initial point cloud into several sub point clouds, at the patch-scale, which can then be successively processed by G3Point. If G3Point can be directly applied to point clouds, without any field constraint on grain size, we generally recommend validating the results against some field measurements (e.g., grain-size distribution obtained by a Wolman count), at least on some parts of the studied area. When no classical grain size data is available, we recommend to carefully check the results of the grain segmentation phase and to test its sensitivity to the different parameters of G3Point. For instance, this could be the case for the automatic measurement of grain size and shape on other planetary bodies (Szabo et al., 2015; Lauretta et al., 2019; Burke et al., 2021) or in inaccessible and remote areas. The outcomes of G3Point are tightly linked to the choice of the local neighborhood scale through the parameter $k$. This parameter should therefore be taken as small as possible, to enable the segmentation of small grains, but not too small to prevent the over-segmentation of large grains due to local topographic minima associated to surface roughness or noise. Suitable values of $k$ are generally determined by a trial-and-error series of tests.

### 4.2 SFM or LiDAR derived point clouds?

As demonstrated in this paper, G3Point can be applied to point clouds obtained with a terrestrial LiDAR or by SFM. Point clouds obtained with LiDAR data provide better accuracy than SFM but can be associated to varying resolution, while the ones obtained by SFM provide uniform resolution but can lead to some inaccuracies. In particular, point clouds obtained with SFM were observed to generate smooth or inaccurate topographic transitions between grains, as these correspond to "shadow" areas difficult to capture with pictures. These smooth transitions are not too problematic for G3Point, as it is based on the steepest slope, but they prevent efficiently using criterion based on topographic curvature to segment grains or to correct the segmentation obtained with G3Point. In that case, we recommend removing points located at local topographic minima to ease segmentation (this is a build-in option). For LiDAR data, the issue of spatially varying resolution can lead to a non-optimal set of parameters, in particular $k$, the number of nearest neighbors considered, over the entire surface of the considered point cloud. In this case, we recommend working on sub point clouds of rather homogeneous spatial point density. The use of point clouds obtained with only one station does not represent an issue for the watershed segmentation of G3Point (Fig. 2), even if




it limits the number of data points per grain and their spatial distribution along the surface of the grains, which is not optimal for shape fitting algorithms. In any case, the point clouds processed by G3Point must be beforehand cleaned of any geometrical feature not corresponding to pebbles. This mostly includes trees, trunks, vegetation, water surface, human-made objects and patches of fine grains (i.e., smaller than the minimal detected grain size).

## 4.3 Comparison of G3Point with previous methods

In terms of total working time, using G3Point over a surface area of about 1-100 m$^2$ captured by SFM involves collecting field pictures (~5-10 min), processing the pictures by SFM to obtain a point cloud (10 min to several hours on a laptop) and running G3Point several times to find a good parametrization (~10 min). Interestingly, G3Point itself is not the limiting factor, as field data acquisition (i.e., pictures or LiDAR data) and data processing (i.e., SFM) appear as more time consuming. This total working time is roughly equivalent to the one of a typical manual pebble count, which takes about 60 min to measure the three axes of 100 grains. However, data sampling for G3Point is not destructive, it can be done by a single operator and G3Point will result in the measurement of a much larger number of grains (>10$^2$ grains) including their size, location, and orientation in 3D. It offers a real benefit in terms of representativity and opens new avenues to quantitatively characterize populations of grains (e.g., not only their size distribution). In fact, because point cloud data acquisition on the field is fast, large areas or multiple locations along a fluvial system can be documented in a limited amount of time. In addition, pictures for SFM can be acquired with drones so that remote locations or very coarse-grained environments can safely be characterized. Together with the large number of grains being considered, G3Point represents a real improvement in terms of spatial representativity with respect to Wolman or photographic approaches which are usually limited to a few squared meters and a hundred of grains (Bunte and Abt, 2001). Last, while most methods based on 3D data use texture or any other morphological index to estimate the grain sizes (Vazquez-Tarrio et al., 2017; Woodget et al., 2018; Chardon et al., 2020), G3Point works directly on the grains and does not require a calibration phase. Once again, this limits bias and time spend on the field and allows remote areas to be characterized.

## 4.4 In situ results on the granulometric conversion factors

Because G3Points samples virtually all the grains at the surface, it belongs to the family of areal or area-by-number grain sampling approaches. To compare this distribution to the Wolman field counts, it must be converted to a grid-by-number distribution, which is considered equivalent to a volumetric grain-size distribution. Conversion factors have been proposed to convert grain-size data acquired with one approach to another one, based on geometrical arguments (Kellerhals and Bray, 1971; Church et al., 1987; Diplas and Fripp, 1992). For example, converting an area-by-number (or areal) distribution to a grid-by-number (or volumetric; e.g., Wolman) distribution requires multiplying the frequency of all the particle classes by a factor D$^2$. However, this exponent of 2 is theoretically valid only for spherical sediments with the same density and without porosity. The use of such conversion factor thus requires a calibration phase and should, in any case, only be considered as an approximate conversion method (Bunte and Abt, 2001).





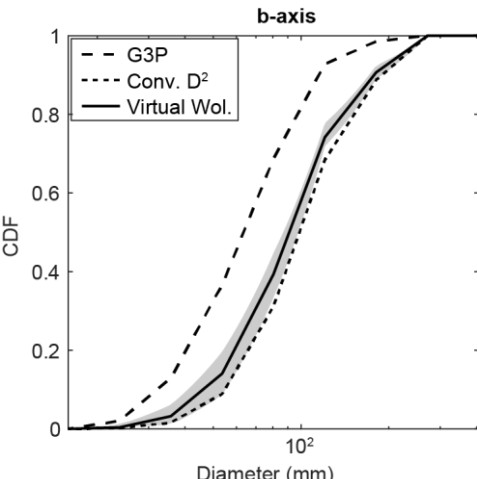

**Figure 7:** Illustration of conversion from a G3Point grain-size distribution to a Wolman-like distribution. Data are from Site 2 of Chateau Renard. The initial G3point distribution is an area-by-number one (large dashed line) that can be converted to a grid-by-number (e.g., Wolman) one with a conversion factor of 2 (small dashed line). Alternatively, a virtual Wolman count can be performed directly on the segmented and fitted grains (black line). The shaded envelop indicates the variability observed with 50 realizations.

With our new approach, we work on 3D point clouds covering large areas and a large number of grains can be identified. Therefore, instead of converting the area-by-number distribution to a grid-by-number one, we can apply a virtual grid over the point cloud and perform a Wolman count on the fitted grains. To account for the spatial variability of the grains, we repeat this operation 50 times to define an uncertainty envelop and use the average distribution as the grain-size distribution of the sample. For our field examples, we observe that the geometrical conversion is always coarser than the virtual Wolman distribution, yet within uncertainties (Figs. 7, S4, S5, S6). The only exception is for the $c$-axis of the grains with the IE fit. Because this fit leads to very flat ellipsoids, the geometrical conversion factor largely overestimates the size of the grains (Figs. S4, S5, S6). In agreement with previous works (Graham et al., 2012), this suggests that the geometrical factors are a correct approximation that tend to maximize the size of the grains, so that Wolman counts should be favored when possible. We emphasize that the field examples presented above were acquired in order to test our approach and the extent of the point clouds are thus similar to the extent of the Wolman counts performed on the field. Therefore, we sampled about the same number of grains on the field and virtually (Table 1). Yet, G3Point is designed to operate on larger point clouds so that a few hundreds of grains will be sampled with the virtual Wolman sampling, allowing for an even more accurate description of the grain-size distribution from point clouds

## 4.5 Opportunities to explore and measure uncharted metrics: grain 3D sphericity and orientation

Here, we briefly present some results on the orientation and sphericity of grains that we obtain with G3Point. The idea is not to dedicate a detailed study of these two metrics, but to illustrate the ability of G3Point to automatically measure them with no




additional efforts. This represents a real benefit of G3Point as most field measurement of grain sphericity and orientation are either cumbersome or approximate (e.g., using qualitative classification).

The azimut and dip angles of a grain may give some information about the flow that transported and deposited a population of grains. G3Point offers a very simple way to access the orientation of a large population of grains as the azimut and dip

angles can easily be determined from the fitted ellipsoids (Fig. 3). On average, the two fitting methods are efficient at recovering orientation, but they do not lead to the exact same results (Fig. 4I). Therefore, if grain orientation is a key element of a study, preliminary tests may be useful to determine the best fitting approach in terms of orientation (which may depend for example on the geometry of studied grains). Here, we show the results of both approaches to illustrate their similary and differences. Azimut is given with respect to the y-axis defined as parallel to the main water flow. At Site 1, the grains show

no preferential azimut (Fig. 8a) and most grains rest flat on the beach, with a dip angle smaller than to 30° or larger than 150° (Fig. 8b). However, 40 to 50 % of the grains exhibit a dip angle between 30° and 150° and are thus quite vertical. We propose that their orientation results from their fall from the very nearby cliffs rather than from transport by the sea. At Site 2, a slightly preferential orientation can be inferred from the DLSF fit, with more grains showing an angle with the main flow than grains aligned with the flow (Fig. 8c). Here again, most grains rest flat and 30-40% of them exhibit a dip angle comprised between

30° and 150° (Fig. 8d). We propose that this is due to a stronger control of the sea on this site with respect to Site 1. Along the Hérault River, grains tend to orient themselves perpendicular to the main flow (Fig. 8e) and to rest flat, with 27-38 % of them with a dip angle comprised between 30° and 150° (Fig. 8f).

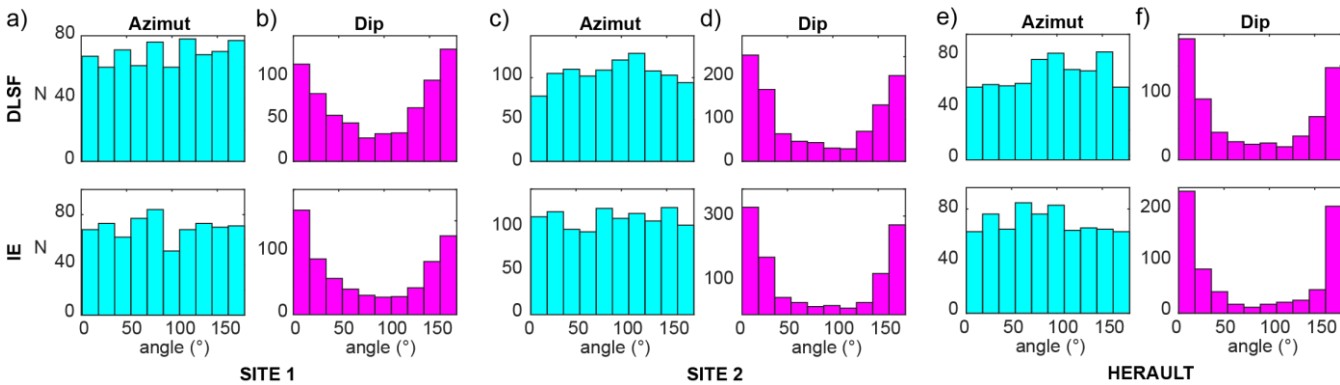

**Figure 8:** Azimut and dip angles of the grains fitted the two approaches (DLSF and IE) at a-b) Site 1, c-d) Site 2 and e-f) the Hérault River.

N is the number of grains of a given angle in degree.

Another potential application of the G3Point is to measure the sphericity of the sediment population at a high level of accuracy as a large number of grains can be considered. Sphericity, $\psi = \left(\frac{bc}{a^2}\right)^{1/3}$, can be interpreted as a proxy for travel distances, when comparing sediments having the same source rock (Bunte and Abt, 2001). A low sphericity (close to 0) is associated to

angular grains and thus suggests a short transport distance. On the contrary, a sphericity close to 1 is associated to smooth grains and suggests a long transport distance. To illustrate this point, we generate 1000 grains from the grain-size distributions





sampled by G3Point (Fig. 5) and calculate the sphericity of the grains. We observe that the grains at Site 1 of Château Renard are associated with a slightly lower sphericity, with a median value of ~0.63, than the grains from Site 2, with a median value of ~0.67 (Figs. 5 and 9). The grains from Site 2 are closer to the shoreline and we propose that the difference in sphericity could reflect their tendency to be more frequently moved during tides as they share the same source rock. The fluvial sediments

from the Hérault River are also associated with a high sphericity which suggests that they are frequently moved, in agreement with qualitative field observations.

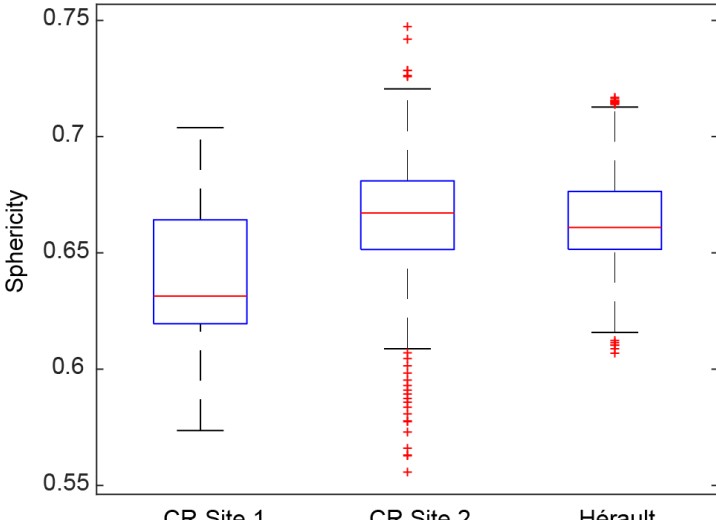

**Figure 9:** Characteristic sphericity of the sediments segmented by G3Point, for the three study sites. One thousand grains are generated from the grain-size distributions obtained with the algorithm (Fig. 5) and their sphericity is calculated using the formula $\psi = \left(\frac{bc}{a^2}\right)^{1/3}$. The red

line indicates the median value, the box represents 50% of the data and 100% of the data are within the whiskers. Red crosses indicate outliers.

## 5 Conclusion

The G3point algorithm presented here solves the issue of grain segmentation and shape analysis from 3D point cloud data. G3Point represents a methodological advance compared to previous granulometric approaches, including hand measurements

or 2D image analysis. Its main advantages are 1) its computational efficiency and speed that relies on the use of a state-of-the-art watershed algorithm (e.g., Braun and Willett, 2013) to segment grains, 2) its scale-free approach which enables to segment grains of large range of sizes above the "neighborhood scale" (i.e., typically a few centimeters), 3) its 3D nature which enables to obtain metrics (e.g., sphericity, orientation) which are seldom obtained in the field, and 4) the large number of measurements, which favors a good representativity of the results.

The G3Point algorithm was able to detect all the grains of a synthetic experiments and to properly describe their size and orientation. It was also qualitatively successful in segmenting hundreds to thousands of grains in fluvial and coastal



environments and to quantitatively capture their size-distribution, compared to hand measurements (e.g., Wolman count). The modelling of grain geometry was performed using ellipsoidal models obtained either with a direct-least square fitting approach or by taking the inertia ellipsoid. If both models lead to accurate inference of the major and intermediate axes, the inertia ellipsoids and the direct-least square ellipsoids tend to underestimate or over-estimate the minor axis, respectively. This in turn impacts the ability of G3Point to infer the volume and surface area of grains. Taking for the minor axis the mean value of the inertia and direct least-square ellipsoids provides estimates that are consistent with hand measurements. Other geometrical models were tested, including bounding boxes. We acknowledge that future works could focus on providing better geometrical models or better fitting approach to describe the geometry of grains.

G3Point is not the first algorithm to propose the segmentation of grains based on point cloud data, as Chen et al. (2020) developed an efficient deep-learning workflow to segment grains based on SFM data. Yet, G3Point is a purely geometric algorithm, which in turn does not rely on the apriori training of a neural network on thousands or more of grains which is required in Chen et al. (2020). G3Point could also represent a good alternative to train deep learning algorithms, as it can provides in a few minutes thousands of grains that otherwise take weeks of work when manually labelled.

Fascinating and first order issues remain to understand the shape and size of grains and interpret them in term of abrasion and fragmentation processes (Domokos et al., 2014, 2015, 2020; Novák-Szabó et al., 2018). This is pivotal for better exploiting the unique geological archives contained in the size, shape and orientation of grains found in natural systems on Earth and other planetary bodies (e.g. Szabo et al., 2015). G3Point, by filling a methodological gap, could foster the development of a more systematic characterization of grain shape in natural environments and lead to a better understanding of the physics of geomorphological processes and of their past dynamics.

**Code availability.**

A MATLAB version of the algorithm can be accessed through a GitHub and/or a Zenodo repository: https://github.com/philippesteer/G3Point/ and https://doi.org/10.5281/zenodo.6368501 (Steer, 2022)

**Author contributions**

PS and LG wrote the paper. PS initiated this study, developed the numerical algorithm, performed the initial tests and provided funding. LG performed the analysis and tested the algorithm in natural environments. PS, LG, DL, AC, AG acquired field data, motivated the study, and contributed to the writing of the paper. All authors checked and revised the text and the figures of the paper and contributed to the ideas developed in this study.

**Competing interests.**

The authors declare that they have no conflict of interest.



**Acknowledgments.**

We thank Thomas Croissant, Edwin Baynes, Benjamin Bruneau, Benjamin Guillaume, Lucas Pelascini, Romy David and Simon Abel for their help acquiring data. This project has received funding from the European Research Council (ERC) under the European Union's Horizon 2020 research and innovation program (grant agreement No 803721). We also acknowledge
support by Université Rennes 1 and CNRS.

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



**Appendix A: The influence of grain surface cover on the resulting ellipsoid size and orientation**

Two strategies are adopted to describe the geometry of a grain with an ellipsoidal model: fitting an ellipsoid by a direct least-square fitting approach (DLSF) or determining its ellipsoid of inertia (IE). We here test the influence of using these two strategies on the quality of the resulting geometrical model considering a variable surface covered by the point cloud (Fig. A1). Indeed, natural grains have a significant proportion of their surface that is not topographically described, as it is hidden under the grain itself or by other grains or features (e.g., vegetation, water) or due to a lack of visibility with respect to the sensor (e.g. LiDAR station). The tested grains consist in a spherical ball (grain 1), a low-angularity grain (grain 2), an angular grain (grain 3) and an angular, flattish and elongated grain (grain 4). The point clouds representing the surface of these four grains were obtained by SFM using Agisoft Metashape.

For each of these point clouds, we generated ellipsoidal models considering only a prescribed percentage of their surface covered by the point cloud, from 10 to 100 %. Practically, surface cover is varied by first choosing a random seed among the points of the point cloud and then sampling a number of nearest neighbors leading to the seeked surface cover of the grain. Ellipsoidal modelling by DLSF and IE is then applied only to this sampled part of the total point cloud.

The modelled ellipsoidal volume $V_{model}$ and surface area $A_{model}$ are then compared to the volume $V_{true}$ and surface area $A_{true}$ of the convex hull of the point cloud. The modelled diameters $d_{model}$ of the 3 axes are compared to the dimensions $d_{true}$ of the bounding box of the point cloud. Last, the 3D angle $\Delta\alpha$, between the modelled orientation of the ellipsoid axes and axes of the "true" ellipsoid obtained by considering the entire grain, is computed. For each surface cover, 10 samples are tested, leading to 10 models obtained by the DLSF and IE approaches, allowing us to define a mean value and a standard deviation for each metric.

For the two low angular grains (grain 1 and 2), metrics obtained with DLSF or IE are consistent with the true geometry of the grain even for relatively low surface cover, down to 20-30%. DLSF gives significantly better results than IE, in particular for a surface cover between 20 and 80%, which likely represents a common range for most labelled grains. Thanks to grain curvature, the DLSF fitting algorithm also converges towards value for $V$, $A$ and $d$ which are close to the true values. For the orientation, both approaches are unable to converge towards the true one for the spherical grain (i.e., grain 1), which is not surprising as the orientation of a sphere is not defined. For grain 2, both approaches converge slowly towards the true orientation for a surface cover greater than 50-75%.

For the angular grain (grain 3), the DLSF and IE approaches give similar results. The dimensions are well captured for a surface cover greater than 60-70 %. The orientation, in particular of the c-axis, converges more rapidly than for low-angular or spherical grains. For the angular, elongated and flattish grain (grain 4), the IE approach gives better results than the DLSF for the length of the c-axis and the volume, while other metrics are relatively similar. Indeed, the algorithm of the DLSF imposes some constraints on the minimum size of the c-axis compared to the a-axis, which makes it unable to properly capture the 3D dimensions of flattish grains.









**Figure A1.** Influence of the grain surface covered by 3D data on the modelled ellipsoidal geometry of a grain. a) Point clouds of the 4 tested grains which consists in grains with increasing angularity and elongation from left (grain 1) to right (grain 4). b) Resulting bounding box (green), and ellipsoids fitted on each grain (black dots), using either the direct least-square fitting algorithm DLSF (red) or the inertia ellipsoid algorithm IE (blue). c) Volume V and d) surface area A of the modelled ellipsoids normalized by the volume and area of the convex hull of the point clouds of the entire grains, considered as true estimates. Length of the modelled e) a-axis, f) b-axis and g) c-axis normalized by the major, intermediate and minor length of the bounding box around the entire grain. 3D angle between the 3D vector of the h) a-axis, i) b-axis and j) c-axis with the orientation of the same vector resulting from the ellipsoid fitting the entire grain. In panel c to j, results obtained with the direct last-square fitting approach (DLSF) and the inertia ellipsoid approach (IE) are represented in red and blue respectively. The error bar, given as a shaded surface around the mean value (solid line), is the standard deviation of the considered metrics obtained by changing ten times the random seed.

These results show that the dimensions of spherical or low-angular grains are well captured by the IE and DLSF approaches, with this latter giving good results even for a surface cover lower than 50%, while their orientation is poorly captured for a surface cover lower than ~75 %. On the other hand, grains that clearly depart from the spherical model, in particular due to their high angularity, need a greater surface cover, around 60-70 %, to be properly captured for their dimensions by ellipsoidal models, while their orientations converge more rapidly. Flattish grains are better modelled by the IE approach, as the DLSF leads to large value of the c-axis. Last, we note that the orientation of the c-axis is generally better captured than the one of the a- and b-axis, which suggests that the azimuthal orientation of grains is less well resolved than their inclination (assuming than the c-axis of grains is sub-vertical).