# Peer review of "Size, shape and orientation matter: fast and semi-automatic measurement of grain geometries from 3D point clouds"

_EGUsphere, 2022_

## Referee Comment (RC1)

**Review of "Size, shape and orientation matter: fast and automatic measurement of grain geometries from 3D point clouds" by Steer et al.**

In their study, Steer et al. present a novel method for point cloud segmentation to retrieve individual pebbles from lidar and SfM point clouds gathered in fluvial environments. The method uses a modified watershed approach, which is common in computer vision segmentation applications, to get grain boundaries. There are several cleaning steps to avoid issues of over-segmentation (single grains being split into many grains) and these must be controlled by a number of parameters and trial-and-error operator tests, with an end result of a qualitatively (to the operator) good segmentation. The choice of model for retrieving the pebble axis dimensions and orientation is explored, and the authors highlight some deficiencies of these models. The method is applied to a number of scenarios in the lab (what they call "synthetic") and field, for which validation data is available, and seems to provide reasonable results for pebble size measurement. Overall, I find the study makes some good progress on pebble segmentation from point clouds, but it needs some significant expansion of the methods to be more explicit and clear, and more discussion of the caveats of the methods (e.g., qualitative segmentation, use of a highly imperfect model for angular grains). I would test the G3P algorithm on some pebble data (the algorithm is published on GitHub, great!), but I am hesitant to do so without a clearer understanding of how the parameters should be set based on point cloud density and other dataset characteristics.

Below, I provide some major comments, followed by a number of line comments (which admittedly sometimes go beyond grammar and citations to verge on major comments), and finish with the important references that I found were missing from the manuscript. I recommend the paper for major revisions before further steps, and would enjoy re-reviewing the paper and testing the algorithm at a later stage.

**Major Comments**

In general, there is some lack of specificity in the methods and the need for some more schematic figures early in the paper showing how grains are segmented and cleaned. Additionally, the authors do not satisfactorily explain how parameters are chosen or how they impact results, especially with regards to more and less angular pebbles, which I imagine is one of the biggest issues with this segmentation approach – not to mention how point cloud density impacts the segmentation. I expand on these concerns below.

In the presentation of the watershed method the authors cite the steepest decent algorithm. Although they make the point that this is usually applied to grids, I did not follow how this algorithm (based on the 8-neighboring grid cells) is the same as the point cloud approach here. If the authors want to talk about flow routing on point clouds, then they should carefully read and cite the FFN method of Rheinwalt et al. (2019). I see the temptation to introduce this in the flow-routing context (especially because of the term "watershed") but I think the connection needs to be elucidated in more detail. Again, this is not the D8 algorithm that is cited, rather it is more of a network-based approach to point cloud segmentation.

Throughout the manuscript the use of "trial-and-error" and "satisfying segmentation" are rather vague. I see that a trained operator who has been to the field site and knows the characteristics of the point cloud data (i.e., collected it themselves) will be required for using this method, and that should be stated explicitly in the abstract and early in the paper. The algorithm needs careful control and likely a lot of tweaking to get "decent" results in a new area. And if this area is large and heterogeneous (in pebble size / shape and point cloud density / quality) it could represent significant effort to get results, which I feel is glossed over in the presentation of the tool. The authors mention the need for validation data at the field cite (i.e., manual Wolman) and that is a significant caveat of the method. If validation data is always needed to "trust" the results, then this needs to be presented up front in the manuscript.

Somewhere in the methods (possibly with reference to the supplement) a schematic figure is needed to show in detail how the "flow" is routed in a neighborhood of pebbles and how the "k" parameter affects this routing. The schematic could also include a cartoon of the three criteria for over-segmentation cleaning, which was not clear based on the description provided. As this is a new method, schematic diagrams (i.e., cartoons, or zoom-in on real data with the points clearly visible) are highly recommended – I would say necessary. This diagram (or diagrams if you put more in the supplement, but I suggest at least one in the main text) should also highlight how the parameters affect the segmentation and cleaning. Alternatively, if this is difficult to show in the diagram, then the parameters should be explicitly treated in a new methods section where the parameters are walked through, with a paragraph or a few sentences for each parameter in which the authors discuss "rules-of-thumb" or quantitative reasons (e.g., point cloud density vs. size of pebbles) for the selection of certain parameters and some trial-and-error scenarios.

As an aside, was under-segmentation never (or rarely) an issue with this method? I suggest the authors read and cite Purinton and Bookhagen (2019, 2021) when discussing over- and under-segmentation, as we go into great detail on these affects for 2D image segmentation (e.g., Section 4.2.1 in the 2021 paper). In the 2019 PebbleCounts paper we discuss specifically watershed segmentation techniques and their tendency to over-segment grains in 2D images (cf. Figure 1, Figure 2, Section 1.2). This is based on 2D segmentation, but I can see how there might be some important parallels to the 3D case (i.e., grain angularity is key!).

I had a look at the Otira_1cm_grains.ply file on GitHub. Is this the exact cloud that was used to develop the watershed model? I wonder, because it looks like this is from a single TLS view and thus has significant occlusions on the back-side of grains facing away from the sensor. I imagine that is really problematic for the model development (based on the issues highlighted in Appendix A). Could the authors please comment on this point cloud? Maybe it is just a subset / sample data provided on GitHub and not the full dataset, which is fine.

Regarding the model choice in Section 2.4, I find myself asking: are pebbles really ellipsoids? I think that is an okay assumption and it is the one that is commonly made (well Domokos et al., 2014 make the case for superellipsoids https://en.wikipedia.org/wiki/Superellipsoid, which I'm not sure I agree with either), but it should be noted that this is a very imperfect model done for the sake of simplicity. The labeled grain in the point cloud could also be directly measured on, but historically we have treated grains as ellipsoids with a/b/c axes. The benefit of the proposed method is that you get an entire grain surface not just three axes. The grain surface isn't totally taken advantage of here (a model is still fit and used to extract those three axes) but this fact should be noted: you get the entire (or sampled at the

point cloud density) grain surface. One path forward could be in new measurements of grain angularity from point clouds and comparing the volume of pebbles (e.g., convex hull approach that the authors use). Maybe mention that with the ability to measure the entire pebble surface in point clouds, going forward we may be able to measure additional (more accurate) grain characteristics, particularly for grains that are far from ellipsoidal shapes, for instance in mountain headwaters or hillslope deposits. But these measurements will certainly depend on the point cloud density and the size of the pebbles we are interested in. I don't think we are at the point of putting numbers on the appropriate density for a given pebble size, and this is highly dependent on pebble shape, but making some statements (in a new discussion section covering all these points?) along these lines would be some great food for thought and place this paper in the context of advancing our understanding of the true shape characteristics of sedimentary deposits. I think a bunch of the citations that wound up in the introduction and conclusion (Domokos, Szabos, Miller) could be nicely woven into these statements. Have a look at the interesting shape characteristics and two phase abrasion model used by Miller et al. (2014) specifically (the authors cite this paper) – how could the G3P method aid in applying this and measuring surface curvature of the pebbles? Very interesting to think about!

Appendix A is really important. I think this needs to be worked up into the main text, probably at the beginning of the results section. See my comments about that later.

**Line Comments**

Beyond the expansion of scientific content, I suggest another careful read-through of the manuscript for additional and/or similar grammatical errors as I point out in my line edits below. I have highlighted a fair number of grammatical errors that caught my eye, but there are still more in there. As a native English speaker, I found these distracting from the scientific content, but they are all easy fixes. I hope the authors take them in good faith!

P1L7: The first sentence of the abstract is a bit awkward with all the "and" statements. I suggest re-writing this to make it more clear. Consider using an oxford comma here and elsewhere, that could make such lists clearer.

P1L11: Comma after "measurements"

P1L14: "into" should be "of"

P1L15: "individualized" should be "segmented"

P1L16-17: "If different…" sentence is awkward and should be rephrased.

P1L20: Point "2)", isn't the limit also the size of grains and point cloud density? This is subtle and you state this as a limit at the end of the abstract. Although there you use the term "point cloud resolution". I prefer "point cloud density" and suggest you use that throughout and mention the density of the point clouds (in pts/m2 or pts/cm2 might make more sense for the high-density SfM datasets). I might remove point 2 from this list anyway.

P2L29: "Collection of a"

P2L30: Add citation for Purinton and Bookhagen (2021). We go into great detail on the issue of sample size there, I suggest the authors have a close look at e.g., Sections 4.3, 5.1, 5.2

P3L1: I think the term is more commonly "photo-sieving"

P3L11: "spent in". Here and elsewhere in the manuscript the phrasing should be "in the field", not "on the field".

P3L26: The full and correct name of what we often call SfM processing is "structure from motion with multi-view stereo (SfM-MVS)" (Smith et al., 2015). This should be stated, and the authors could say something like "structure from motion with multi-view stereo, herein referred to as SfM". I think the lowercase "f" is more common.

P4L6: "oriented"

P4L7: "reorient"

P4L10: Here and elsewhere "point density", not "resolution"

P4L17: This is commonly called "D8", please mention this. The Facet Flow Network (FFN) algorithm of Rheinwalt et al. (2019) should certainly be carefully read and cited if you are discussing flow routing on point clouds (cf. https://github.com/UP-RS-ESP/FacetFlowNetwork).

P5L3: Why is k=20 here?

P5L10: "approach is as"

P5L18: The first sentence in Section 2.2 should be broken into multiple sentences. The authors should elaborate on attempts made with different clustering approaches. I agree these are tricky, but be explicit with what methods were tried and possibly what the issues / strengths / weaknesses were of the results.

P5L19: I think "DBSCAN" gets capitalized.

P5L29: I don't understand how the "drainage area" relates to the grain radius. Might need more explanation here, or highlight this in the schematic diagram I proposed.

P5L29-30: Cf seems like a tricky parameter and will have a big effect on the segmentation. I really don't get where this is coming from. Basically, it seems like the segmentation needs to be carefully checked and modified by a trained operator. This should be made clear up front.

P5L32: missing period

P6L2: is k=20 here as well?

P6L3: "between": which normals are compared? Is it the nearest normal in the i and j grain? This would help to show diagrammatically

P6L5: "equals" should be "equal"

P6L6: "60 degrees throughout the paper". But why 60? And this is a free parameter so I guess not "throughout the paper". But I don't understand "in the following" here. "Following" where?

P6L6: Curved how? I'm having a lot of trouble understanding this criterion. I think all of these criteria need a schematic figure.

P6L7: Delete "Therefore"

P6L19: the objective here is to merge small and large grains?

P6L20: "This number of …" don't understand. greater than 10? greater than or equal to k? what is k? is k=20?

P6L24: I guess this is another step that needs very close operator control and field knowledge. I see the algorithm works quickly, but I imagine that this requires a lot of going back and tweaking parameters until a "satisfying" (qualitative) segmentation is reached. This probably takes a while even on a small patch with <1,000 grains as here, and would be difficult to upscale to large areas or to finer point cloud spacing (i.e., SfM at hundreds to thousands of points per cm2) where you would be able to get smaller grains and thus exponentially more segmentation would be performed (log-normal GSD typical of river sediments). SfM is used later on, but this point about the extremely high density of SfM point clouds and the exponential increase in smaller grains is never made? Or I may have missed it.

P6L27-28: "respectively" is maybe missing, if you are referring to the beginning of the sentence.

P6L32: Why wasn't the example in Figure 2 cleaned? It wasn't required because the segmentation worked so well?

Figure 2 caption: "fit" not "fitted", here and elsewhere in the main text. "same color as in panel c". "relative" not "relatively"

P8L6: "consists of"

P8L7: Don't understand the use of "adequacy"

P8L9-10: "From directly using the point cloud to describe each grain and measure their sizes and orientations."

P8L10: Delete "the use of"

P8L22: Delete "Anyhow", too informal

P8L28: It's not really possible to see anything in Figure 2. You may need another figure with a zoom-in on a couple of grains (the original points, colored by grain label) and the ellipse fit to the points. This may be hard to visualize, but as it stands you cannot see what is being referred to here in Figure 2. cf. Appendix Figure A1. This is going in the direction of a useful figure in panel b, but the points should be increased in size and the model fits made semi-transparent. I realize these figures are screenshots from cloud compare, but I wonder if they could still be somewhat improved by zooming in more and increasing the point size / adjusting the colors?

P8L29: "best solution". What specifically was tested besides the least squares?

P8L30: Could you provide a reliable citation for "inertia ellipsoids"? I was doing some internet searching and having trouble understanding what these are and how they are computed. I don't get the relation

between the equation of an ellipsoid and this model / mathematical operation. Even an internet link would help me here! I just don't get it, but maybe I need to accept it as mathematical fact.

P8L33: I think singular value decomposition was already defined. Fine to use "SVD"

P9L5: how were the "ellipsoids" and "cuboids" compared? you mean the dimensions of the cuboid (length / width / height) with the axes of the ellipsoids (a /b /c)? Please be explicit here.

P10L3: "axes"

P10L5: "approximates"

P10L11: "metric"

P10L20: "diameters"

P10L27: I like the lab experiment, with the pebbles from the hardware store, but I wouldn't call this "synthetic". Rather "lab experiment" is more appropriate. Or you say "Lab Environment" and "Natural Environment". Please remove the term synthetic from the manuscript. Synthetic implies artificially generated, but these are real pebbles you are laying out and measuring (in a lab setting).

Figure 4: The cuboid volume is definitely way far off from the true volume, but putting it on the x-axis of panel g here implied this is validation data. If you really wanted to get the volume you could do a convex hull of the labelled point cloud and directly measure the volume from this as a control to compare the ellipsoid volume with. Actually, it looks like you do exactly this in Appendix A! Another reason to put Appendix A at the beginning of the results.

Figure 4 caption: "top", "middle", "bottom" should be "left, center, right".

P11L11: This experiment does not test the watershed segmentation at all since the pebbles are not overlapping. Or at least, this is barely a test. That's important. The first experiment is really ONLY for testing the ellipsoid models and axis length measurements as far as I can tell. Be explicit about this point.

P11L12: "captured by pictures" should be "photographed"

P11L13: "Processed" how? Be specific. High resolution? How many interest points? Was any filtering done? What was the resolution of the input images and what camera model was used to take the photos? How many oblique vs. nadir photos and approximately how low of an oblique angle was used?

P11L14: How as the planar surface removed? Manual cropping in CC? If this surface is removed, then the watershed segmentation is really barely necessary, so again this isn't testing the segmentation algorithm -- just the model fits.

P12L10: I wouldn't call these consistent. Maybe it helps to include a metric like RMSE of the two methods compared to the cuboid and plot this number (or two numbers) on each of the subplots in Figure 4e. It actually looks like the bias increases at larger grain sizes, why is that?

P12L15: The discussion of the DLSF and IE models is really interesting. I wonder if this is better highlighted in the Appendix A section, which I recommend becoming a new section at the beginning of the results.

P12L20: I'm not sure about using cuboid volume in any comparisons, this is not a volume of the grain (actually it's really far from the volume of a grain contained in the bounding box).

Figure 5: Struggling with these CDFs. Maybe make the G3P lines dashed in the plot and legend?

Figure 5 caption: "envelopes"

P14L1: So this is equivalent to the cuboids used in the first experiment? May be interesting to note whether the cuboid dimensions did in fact correspond well with manually measured dimensions of the hardware-store bought pebbles. No need for a formal analysis, but was this checked ever? Do the cuboids accurately give the hand-measured lengths?

P14L2: "wo"?

P14L5: Why no a and c axes here? Time constraint? Just curious.

P14L9: This is a fair approach, but it would also be nice here to cite the binomial modeling approach of Eaton et al. (2019) and its application in Purinton and Bookhagen (2021).

Table 1: Thanks for noting the final parameters, but what were the steps in their selection (what other values were tried?). Is this just something the operator needs to "get a feel" for? Or are there good reasons for these values? This goes towards one of my major comments regarding the parameters.

P14L19: What is the MP resolution of the images?

P14L19: Here and elsewhere in the manuscript it should be "square meters" not "squared meters".

P14L23: "trial-and-error" here and elsewhere is vague. Can you be more specific about how these tests were run? How were initial values selected and how were the modified based on the runs? Was there systematic adjustment of the parameters or was it a bit more "random"?

P15L1: "grains are detected"

P15L5: Again, incorrect use of "synthetic". You are measuring real grains on the point cloud.

P15L13: "percent"

Figure 6: Why only considering these percentiles? Maybe It would be cool to have a QQ plot where you show several percentiles plotted against each other for the Wol vs. G3P, with one QQ plot for each site and for each axis (so a 3x3 figure with 9 QQ plots, well 7 plots actually because Herault is missing a and c). Then you could plot e.g., the 5, 16, 25, 50, 75, 84, 95 percentiles against each other in each plot (common grain size percentiles). Could also include the common bias and accuracy metrics on the plot (cf. Purinton and Bookhagen 2019, Figure 12). For each percentile the uncertainty could also be visualized with vertical and horizontal error bars. Would be useful and you could drop Table 2.

P15L22: "quartile" should be "quantile" here and elsewhere. Quartile refers to the 25th, 50th, 75th, and 100th percentiles. Alternative to using quantile, consider just using the term "percentile" which is more common in grain-size studies. Quantile is usually the 0.1, 0.2, 0.3, etc., whereas percentile is 10, 20, 30.

Table 2: I suggest to remove this. A Figure with vertical and horizontal error bars as I suggest in place of Figure 6 would be much easier to read and present the results more strongly.

P17L5: Did you mean "point/cm2"? is this supposed to be a point density? or is this a distance between points?

P17L6: "10^6". But the SfM datasets had way more points right? On the order of 10^7-9 is what I would expect from SfM point clouds of pebbles. I saw you reduced the number to "speed processing", how long would the processing have been at full resolution? Days?

P17L19-20: "Suitable values…" A bit more guidance should be provided. what would be interesting is to show (maybe in the supplement) a zoomed in area of the point cloud with labels resulting from e.g., k = 10, 20, 30, 40, 50, to see how this effects the segmentation.

Section 4.2: By reducing the density of SfM points you are removing one of the key benefits of this method: denser point clouds! That should be noted. Point density could be very useful for measuring smaller grains…

P17L24-26: Yes! we have noted this too on SfM point clouds. This may be a Metashape issue, or an issue with the number of photos and the angle of the photos. It should be noted that the source of these issues may be related to the quality of photos taken (lighting, resolution, blurriness) and/or lack of sufficient coverage / view angles -- it is not necessarily a result of the underlying SfM algorithms, but we need to explore this more.

P17L29: "built-in"

P18L4: Be consistent of your use of commas with "e.g." and "i.e.", in other places you do not use a comma after.

P18L12: "> 10^2"… well that depends on the size of the grains versus size of surveyed areas and the density of the point cloud. It's tricky to put a number on this.

P18L13: Here and elsewhere I think you mean "representation", not "representativity" (not a word).

P18L14: Yes! Here you could mention that an entire surface of the grain is retrieved, so we are not limited to fitting ellipsoid models (though this is still useful w.r.t. historical approaches).

P18L16: Well, drone usage in this case is really challenging, and maybe not realistic, see Section 5.4 in Purinton and Bookhagen 2019.

P18L18: "hundred grains"

P18L21: "spent in"

Section 4.5: Good that you make some of these statements, but I think they can be teased earlier. You are not just limited to the a/b/c axes when we start labeling point clouds, but a wealth of other more accurate information about the grain shape (cite Domokos' work here).

P19L22: Note, orientation is also provided by 2D image segmentation methods (e.g., PebbleCounts), although this is only the "azimuth" and not the "dip".

P20L26: I think "select" not "generate". Why only 1000 grains? Why not use all grains?

Figure 9 caption (and Figure S7 caption): Check your definition of the boxplot. If 100% of the data was always in the whiskers there would be no red outliers outside the whiskers.

P21L13: "solves the issue" is a little strong here. I would say "makes progress on the issue".

P21L14: I think "methodological advance" is also a little strong. Rather, the authors take the watershed approach concept used by many other algorithms / studies and apply it to point clouds in the framework of a network-based approach. This is an "alternative" and a "unique application to point clouds", but I would refrain from "advance" w.r.t. other studies that use different approaches.

P21L19: Not sure what is meant by this last statement, maybe rephrase.

P21L20: Yes, it detected all grains in the lab experiment, but is that an interesting or notable result? The grains were far from each other and not overlapping (and the area between them was removed manually I think?). The lab experiment was more about testing the ellipse models.

P22L7-8: Shouldn't this statement be in the discussion?

P22L9-13: This is really late to be introducing previous and highly relevant work. A discussion of this alternative approach should be up in the introduction or methods section. I agree with the point about G3P used to generate training data, but that could go in the discussion.

Appendix:

P29L1: This appendix is enormously important! I think it should really be a section at the beginning of the results. It really helps my understanding of the models and the limitations based on grain shape. Grain shape is extremely important if you also make the statement about the ellipsoid model being an imperfect descriptor of natural fluvial sediments

P29L7: "consist of"

P29L9: Be specific, how were all sides of the grains collected? Was the object rotated to get a photo from every side? were they laid on a flat surface (so there is still part of the grain "missing")?

P29L10: I'm not sure if you just want a percentage or a rather what is more interesting is taking a percentage from a certain region, or dropping an area of points to simulate an occlusion. Is that what is done? Not clear from this description.

P29L12: "seeked"?

P29L15: Convex hull is used here! It should also be used in Figure 4 rather than "cuboid volume".

Figure A1: Figure is pretty low-resolution. It would be nice if when I zoom in on the panel b I can see the points and ellipsoids, but the resolution is too low for that right now. Not strictly necessary, but this would be really helpful.

P30L14: But these ellipsoidal models are likely increasingly "wrong" for more angular grains. I don't think you need to modify the analysis but this point should definitely be noted! I think the path forward for

grain-size measurement from point clouds does not lie in ellipsoidal models, but rather measurements directly on the labelled points.

P30L15-16: This is important! DLSF overestimates c. We see that in Figure 4. That's why this appendix / figure should come at the beginning of the results.

P30L17: "assuming that"

**References**

Purinton, B., & Bookhagen, B. (2021). Tracking downstream variability in large grain-size distributions in the south-central Andes. Journal of Geophysical Research: Earth Surface, 126, e2021JF006260. https://doi.org/10.1029/2021JF006260

Rheinwalt, A., Goswami, B., & Bookhagen, B. (2019). A network-based flow accumulation algorithm for point clouds: Facet-Flow Networks (FFNs). Journal of Geophysical Research: Earth Surface, 124. 2013–2033 https://doi.org/10.1029/2018JF004827

Smith, M., Carrivick, J., and Quincey, D.: Structure from motion photogrammetry in physical geography, Prog. Phys. Geog., 40, 247–275, https://doi.org/10.1177/0309133315615805, 2015

---

## Author Comment (AC1)

Dear Editor,

Your will find below my responses to the comments made by Benjamin Purinton and an anonymous reviewer. I am very grateful for their positive and constructive comments. Their comments appear in black below while my responses are in blue. The changes made to the manuscript are first detailed after each response (when appropriate) and then appear on the manuscript with the track of changes. For clarity, we have regrouped the summary of the reviews of the two reviewers, their main comments (which sometimes were similar) and then their line or specific comments in 3 different sections.

Best regards,

Philippe Steer and Laure Guerit, on behalf of the co-authors of this manuscript

**REVIEW SUMMARY**

**Editor response by Rebecca Hodge**

Thanks to the reviewers for their comments, which identify a number of important points. Making sure that the details of these methods are clear will be important to encourage others to try these techniques. I encourage the authors to make sure that they respond thoroughly to these reviews.

**Review summary RC1 by Benjamin Purinton**

In their study, Steer et al. present a novel method for point cloud segmentation to retrieve individual pebbles from lidar and SfM point clouds gathered in fluvial environments. The method uses a modified watershed approach, which is common in computer vision segmentation applications, to get grain boundaries. There are several cleaning steps to avoid issues of over-segmentation (single grains being split into many grains) and these must be controlled by a number of parameters and trial-and-error operator tests, with an end result of a qualitatively (to the operator) good segmentation. The choice of model for retrieving the pebble axis dimensions and orientation is explored, and the authors highlight some deficiencies of these models. The method is applied to a number of scenarios in the lab (what they call "synthetic") and field, for which validation data is available, and seems to provide reasonable results for pebble size measurement. Overall, I find the study makes some good progress on pebble segmentation from point clouds, but it needs some significant expansion of the methods to be more explicit and clear, and more discussion of the caveats of the methods (e.g., qualitative segmentation, use of a highly imperfect model for angular grains). I would test the G3P algorithm on some pebble data (the algorithm is published on GitHub, great!), but I am hesitant to do so without a clearer understanding of how the parameters should be set based on point cloud density and other dataset characteristics.

Below, I provide some major comments, followed by a number of line comments (which admittedly sometimes go beyond grammar and citations to verge on major comments), and finish with the important references that I found were missing from the manuscript. I recommend the paper for major revisions before further steps, and would enjoy re-reviewing the paper and testing the algorithm at a later stage.

**Review summary RC2 by Anonymous Referee #2**

This study proposed an approach to segment grains in 3D point clouds. Based on bare-terrain point clouds that only include grains and topographic flats, the authors initially utilized a watershed method to roughly segment grains. The initial segmentation sometimes resulted in an over-segmentation problem where some grains were falsely split to multiple parts. Therefore, they conducted three criteria to merge grains. Several operations cleaned noise points and detected topographic flats in point clouds. Two methods approximated ellipsoids from the segmented grains. They also presented histograms of geometric characteristics of the approximated ellipsoids. The first experiment examined the 3D grain segmentation method in synthetic data. They also tested their method at three field sites and compared the results with the Wolman method. The 3D grain segmentation method from this study is interesting and straightforward. Based on the two major assumptions mentioned in the paper, the method seems effective and novel. This is an interesting and promising paper. The reviewer suggests the authors address the comments, rewrite the paper with formal, scientific language, and resubmit it.

We are very grateful to the two reviewers and to the associate editor and editor for their work on our manuscript, for their detailed and constructive comments and for their overall positive evaluation. We below provide answers to their comments. We have in turn made significant changes to the manuscript to increase its quality following the recommendations of the reviewers.

**MAIN COMMENTS**

**Comment 1 (Benjamin Purinton) - Make schemes and explanations on methods and parameter choice**

In general, there is some lack of specificity in the methods and the need for some more schematic figures early in the paper showing how grains are segmented and cleaned. Additionally, the authors do not satisfactorily explain how parameters are chosen or how they impact results, especially with regards to more and less angular pebbles, which I imagine is one of the biggest issues with this segmentation approach – not to mention how point cloud density impacts the segmentation. I expand on these concerns below.

Somewhere in the methods (possibly with reference to the supplement) a schematic figure is needed to show in detail how the "flow" is routed in a neighborhood of pebbles and how the "k" parameter affects this routing. The schematic could also include a cartoon of the three criteria for over-segmentation cleaning, which was not clear based on the description provided. As this is a new method, schematic diagrams (i.e., cartoons, or zoom-in on real data with the points clearly visible) are highly recommended – I would say necessary. This diagram (or diagrams if you put more in the supplement, but I suggest at least one in the main text) should also highlight how the parameters affect the segmentation and cleaning. Alternatively, if this is difficult to show in the diagram, then the parameters should be explicitly treated in a new methods section where the parameters are walked through, with a paragraph or a few sentences for each parameter in which the authors discuss "rules-of-thumb" or quantitative reasons (e.g., point cloud density vs. size of pebbles) for the selection of certain parameters and some trial-and-error scenarios.

We believe the Methods section is already quite dense and sufficiently described. However, we also fully agree with the reviewer that some additional schemes on the different parts of the method workflow associated with some additional explanations on how to choose the parameters are required.

➢ Change to the Manuscript: We have added a significant section in the Supplement Material to better illustrate the methods in a practical way (including Fig. S8, S9 and S10) and providing some sensitivity analysis for each parameter. Moreover, we have added a Figure in the Appendix (Fig. A1) which is based on schematic sketches illustrating the meaning and role of each parameter.

**Comment 2 (Benjamin Purinton) – over- and under-segmentation**

As an aside, was under-segmentation never (or rarely) an issue with this method? I suggest the authors read and cite Purinton and Bookhagen (2019, 2021) when discussing over- and under-segmentation, as we go into great detail on these affects for 2D image segmentation (e.g., Section 4.2.1 in the 2021 paper). In the 2019 PebbleCounts paper we discuss specifically watershed segmentation techniques and their tendency to over-segment grains in 2D images (cf. Figure 1, Figure 2, Section 1.2). This is based on 2D segmentation, but I can see how there might be some important parallels to the 3D case (i.e., grain angularity is key!).

The problem of over or under-segmentation is a classical problem which impact most (all?) segmentation approaches, including grain segmentation in 2D (Purinton et al., 2019, 2021) and in 3D (this manuscript). G3Point is based on the idea that most grains are in any case over-segmented after the initial watershed segmentation. This over-segmentation results from the (already mentioned) fact that "grains can exhibit several local maxima, due to the geometry of the grain (i.e., angularity) or to a rough surface or to potential data noise" (page 7 lines 10-15). Therefore, we have also developed algorithms to correct from over-segmentation (section 2.2.)  Even if this is less likely, under-segmentation can also be an issue when using G3Point, in particular for small grains. This happens mostly when the k parameter (i.e., the number of neighbors) is too large to allow the identification of these small grains. If their number of points is lower or of the same order than the k-parameter, then these grains would be merged with other neighbor grains.

➢ Change to the Manuscript: We have added some sentences in the second paragraph of Section 2.1 to mention under-segmentation and to better address over-segmentation in relation to previous 2D approaches.

**Comment 3 (Benjamin Purinton) - the watershed algorithm** - In the presentation of the watershed method the authors cite the steepest decent algorithm. Although they make the point that this is usually applied to grids, I did not follow how this algorithm (based on the 8-neighboring grid cells) is the same as the point cloud approach here. If the authors want to talk about flow routing on point clouds, then they should carefully read and cite the FFN method of Rheinwalt et al. (2019). I see the temptation to introduce this in the flow-routing context (especially because of the term "watershed") but I think the connection needs to be elucidated in more detail. Again, this is not the D8 algorithm that is cited, rather it is more of a network-based approach to point cloud segmentation.

Please first note that we do not cite the D8 algorithm but the steepest descent algorithm. The steepest descent algorithm, often referred to as the D8 algorithm when applied to regular grids, is a classical algorithm which has been used to route water not only on regular grids but also on irregular grids at least since Braun & Sambridge (1999). As already mentioned in our manuscript, the main difference between the regular- and the irregular-grid version of this algorithm, with respect to the aim of segmentation, is the definition of the neighborhood attributed to each node. For efficiency, we use a

definition of the neighborhood that is simply the k-nearest neighbors, a classical and efficient way to determine neighbors on irregular grids such as point clouds (Braun & Sambridge (1999) uses natural neighbors). The steepest descent algorithm therefore searches for the steepest slope among these k-neighbors, instead of searching among 8 neighbors as in the D8 version for regular grids. Please note that we do not intend to accurately model hydrological flow over the surface of the point cloud, as it is done in Rheinwalt et al. (2019), but simply to quickly segment the point clouds into watersheds. TIN-based methods to determine neighborhood and flow topology are generally more costly in term of computational resources due to the need of triangulating the point cloud. However, they also offer a better description of flow by correctly accounting for the actual surface area covered by each point/node.

➢ Change to the Manuscript: To clarify this last point, we have added a sentence in section 2.1 "We yet emphasize that this algorithm is not intended to provide an accurate description of hydrological flow over a point cloud as in Rheinwalt et al. (2019), but simply to provide a fast segmentation of the point cloud".

**Comment 4 (Benjamin Purinton & Anonymous Reviewer #2) - automatic vs trial-and-error or semi-automatic**

Benjamin Purinton: Throughout the manuscript the use of "trial-and-error" and "satisfying segmentation" are rather vague. I see that a trained operator who has been to the field site and knows the characteristics of the point cloud data (i.e., collected it themselves) will be required for using this method, and that should be stated explicitly in the abstract and early in the paper. The algorithm needs careful control and likely a lot of tweaking to get "decent" results in a new area. And if this area is large and heterogeneous (in pebble size / shape and point cloud density / quality) it could represent significant effort to get results, which I feel is glossed over in the presentation of the tool. The authors mention the need for validation data at the field cite (i.e., manual Wolman) and that is a significant caveat of the method. If validation data is always needed to "trust" the results, then this needs to be presented up front in the manuscript.

Anonymous Reviewer #2: One criticism of the proposed method is that it requires a trial-and-error step to decide the parameters, and the rubric of such exploration is unclear. This is another biggest concern of this study, because it is unknown when to stop such a trial-and-error step to get the best results. If the exploration is just based on visual results, the effectiveness and accuracy of the proposed method is doubtful. Additionally, if the proposed method needs to manually explore and test parameters, it should belong to semi-automatic methods. The authors should clarify these points.

The two reviewers agree that our manuscript does not clearly state that the choice of parameters needed to obtain a good segmentation of the point cloud into grains require a trial-and-error approach, which in turn qualifies G3Point as a semi-automatic method rather than an automatic one (point 1). Moreover, they question the validation of the trial-and-error approach by a visual inspection, when no independent dataset on grain-size (e.g., hand measurements) is available (point 2).

➢ Change to the Manuscript:
For point 1, we have systematically replaced the word "automatic" by "semi-automatic" when referring to G3Point (including in the title). We now state in the abstract: "G3Point is a semi-automatic approach as it requires to determine by a trial-and-error approach the best

combinations of parameter values." We also clarify this point at the end of the introduction. We also provide some guidelines on how to choose the value of each parameter in the Supplementary Material.

For Point 2, a validation of the results obtained with G3Point is always required, as in any approach based on models. We now state in the abstract "Validation of the obtained results is performed either by comparing the obtained size distribution to independent measurements (e.g., hand measurements) or by visually inspecting the quality of the segmented grains." We also clarify this point at the end of the introduction.

**Comment 5 (Benjamin Purinton) - the Otira point cloud** - I had a look at the Otira_1cm_grains.ply file on GitHub. Is this the exact cloud that was used to develop the watershed model? I wonder, because it looks like this is from a single TLS view and thus has significant occlusions on the back-side of grains facing away from the sensor. I imagine that is really problematic for the model development (based on the issues highlighted in Appendix A). Could the authors please comment on this point cloud? Maybe it is just a subset / sample data provided on GitHub and not the full dataset, which is fine.

Indeed, the Otira point cloud was obtained from a single scan with a terrestrial LiDAR (as already mentioned in the first paragraph of the method). We use this point cloud to test the ability of the model to segment grains despite a limited amount of data and a single view (i.e., this could be considered as a problematic scenario). We agree that the dimensions of the grains obtained with this point cloud should be taken with caution, despite a good qualitative agreement (visually evaluated). We note that we have deliberately not compared the size distribution obtained with G3Point with hand measurements in this case. We also use other types of point clouds in the manuscript (e.g., single view or not, TLS or SfM) to highlight the ability of G3Point to handle different sources of point clouds.

➢ Change to the Manuscript: To clarify this last point, we have added two sentences at the end of the first paragraph of the Methods section: "Because it was acquired after a single scan, this point cloud is not optimal to obtain robust information on grain size. However, it represents a valuable test to check the ability of G3Point to detect grains despite this main disadvantage.

**Comment 6 (Benjamin Purinton) – about grain shape and how it deviates from ellipsoids** - Regarding the model choice in Section 2.4, I find myself asking: are pebbles really ellipsoids? I think that is an okay assumption and it is the one that is commonly made (well Domokos et al., 2014 make the case for superellipsoids https://en.wikipedia.org/wiki/Superellipsoid, which I'm not sure I agree with either), but it should be noted that this is a very imperfect model done for the sake of simplicity. The labeled grain in the point cloud could also be directly measured on, but historically we have treated grains as ellipsoids with a/b/c axes. The benefit of the proposed method is that you get an entire grain surface not just three axes. The grain surface isn't totally taken advantage of here (a model is still fit and used to extract those three axes) but this fact should be noted: you get the entire (or sampled at the point cloud density) grain surface. One path forward could be in new measurements of grain angularity from point clouds and comparing the volume of pebbles (e.g., convex hull approach that the authors use). Maybe mention that with the ability to measure the entire pebble surface in point clouds, going forward we may be able to measure additional (more accurate) grain characteristics, particularly for grains that are far from ellipsoidal shapes, for instance in mountain headwaters or hillslope deposits. But these measurements will certainly depend on the point cloud density and the size of the pebbles we are interested in. I don't think we are at the point of putting numbers on the appropriate density for a given

pebble size, and this is highly dependent on pebble shape, but making some statements (in a new discussion section covering all these points?) along these lines would be some great food for thought and place this paper in the context of advancing our understanding of the true shape characteristics of sedimentary deposits. I think a bunch of the citations that wound up in the introduction and conclusion (Domokos, Szabos, Miller) could be nicely woven into these statements. Have a look at the interesting shape characteristics and two phase abrasion model used by Miller et al. (2014) specifically (the authors cite this paper) – how could the G3P method aid in applying this and measuring surface curvature of the pebbles? Very interesting to think about!

We agree with this comment that going beyond ellipsoidal models is the long-term goal and that this is indeed a very interesting perspective. This is also something we would like to test in a future study. However, this is also a relatively challenging goal, and the main limit remains that only a fraction of the surface cover of grains is visible. As this is not (yet) something we explore in this manuscript, adding a discussion section on this subject seems premature. We prefer mentioning this possibility (of extracting information on grain geometry which does not rely on standard 3D geometrical models) in conclusion.

➢ Change to the Manuscript: We have added sentences in the conclusion on this possibility.

**Comment 7 (Benjamin Purinton) – Make Appendix A a result section**

Appendix A is really important. I think this needs to be worked up into the main text, probably at the beginning of the results section. See my comments about that later.

We agree with this comment (this is something we also previously considered).

➢ Change to the Manuscript: We have move Appendix A (including former Figure A1) to the Results section (section 3.1).

**Comment 8 (Anonymous Reviewer #2)– Improve English**

However, English writing is one of the biggest concerns of publishing this paper on ESurf. The paper is full of ambiguous verbs and redundant sentences such that the reviewer has to guess what the authors intend to convey. There are also many transition words misused in the paper.

➢ Change to the Manuscript: The paper has been read and corrected, when necessary, by a native English speaker.

**Comment 9 (Anonymous Reviewer #2)– Improve state-of-the-art**

This paper also does not provide enough research on the previous work about granulometry such as traditional granulometry and recent optical granulometry based on machine learning. Reviewing these studies and comparing them with the proposed method are important to illustrate the novelty or advantages of this study. Unfortunately, the literature review is limited. For example, watershed has already been widely applied to 2D grain segmentation in Bulter et al., 2001. Also, Garbonneau et al. (2018) and Langhammer et al. (2017) are recent machine learning based approaches for optical granulometry. The review of these important studies is missing in this paper.

Existing methods for grain-size measurements and their limitations are presented in the second and third paragraphs of the Introduction.

➤ Change to the Manuscript: Yet, to better acknowledge previous works, we have added the references mentioned by the reviewers, that were indeed missing. We also add a sentence to highlight grain-size methods based on deep learning methods (the papers were already cited but were mixed with other methods).

**LINE OR SPECIFIC COMMENTS**

**Line or Specific Comments (Benjamin Purinton)**

Beyond the expansion of scientific content, I suggest another careful read-through of the manuscript for additional and/or similar grammatical errors as I point out in my line edits below. I have highlighted a fair number of grammatical errors that caught my eye, but there are still more in there. As a native English speaker, I found these distracting from the scientific content, but they are all easy fixes. I hope the authors take them in good faith!

P1L7: The first sentence of the abstract is a bit awkward with all the "and" statements. I suggest re-writing this to make it more clear. Consider using an oxford comma here and elsewhere, that could make such lists clearer.

Done

P1L11: Comma after "measurements"

Done

P1L14: "into" should be "of"

Done

P1L15: "individualized" should be "segmented"

Done

P1L16-17: "If different…" sentence is awkward and should be rephrased.

Done

P1L20: Point "2)", isn't the limit also the size of grains and point cloud density? This is subtle and you state this as a limit at the end of the abstract. Although there you use the term "point cloud resolution". I prefer "point cloud density" and suggest you use that throughout and mention the density of the point clouds (in pts/m2 or pts/cm2 might make more sense for the high-density SfM datasets). I might remove point 2 from this list anyway.

Point 2 is indeed limited by the size of grains respectively to the point cloud resolution. This is what the following sentence of the abstract mentions. We disagree with the systematic use of point cloud "density" instead of "resolution", as resolution has a unit of a length while point cloud density has a unit that is a number of point per area. In this sentence, resolution is the correct word as we are comparing with the characteristic length of grains.

P2L29: "Collection of a"

Done

P2L30: Add citation for Purinton and Bookhagen (2021). We go into great detail on the issue of sample size there, I suggest the authors have a close look at e.g., Sections 4.3, 5.1, 5.2

Done

P3L1: I think the term is more commonly "photo-sieving"

Done

P3L11: "spent in". Here and elsewhere in the manuscript the phrasing should be "in the field", not "on the field".

Done

P3L26: The full and correct name of what we often call SfM processing is "structure from motion with multi-view stereo (SfM-MVS)" (Smith et al., 2015). This should be stated, and the authors could say something like "structure from motion with multi-view stereo, herein referred to as SfM". I think the lowercase "f" is more common.

Done

P4L6: "oriented"

Done

P4L7: "reorient"

Done

P4L10: Here and elsewhere "point density", not "resolution"

Done (where suitable)

P4L17: This is commonly called "D8", please mention this. The Facet Flow Network (FFN) algorithm of Rheinwalt et al. (2019) should certainly be carefully read and cited if you are discussing flow routing on point clouds (cf. https://github.com/UP-RS-ESP/FacetFlowNetwork).

See our response to the main comment 3

P5L3: Why is k=20 here?

We now state that this value provides a good solution to grain segmentation

P5L10: "approach is as"

Corrected

P5L18: The first sentence in Section 2.2 should be broken into multiple sentences. The authors should elaborate on attempts made with different clustering approaches. I agree these are tricky, but be explicit with what methods were tried and possibly what the issues / strengths / weaknesses were of the results.

Changed

P5L19: I think "DBSCAN" gets capitalized.

Done

P5L29: I don't understand how the "drainage area" relates to the grain radius. Might need more explanation here, or highlight this in the schematic diagram I proposed.

Assuming a round shape, the characteristic length or radius of a grain can be defined geometrically based on the area of the grain (which is the drainage area at the summit/outlet of the grain).

P5L29-30: Cf seems like a tricky parameter and will have a big effect on the segmentation. I really don't get where this is coming from. Basically, it seems like the segmentation needs to be carefully checked and modified by a trained operator. This should be made clear up front.

See response to the main comment 1

P5L32: missing period

Done

P6L2: is k=20 here as well?

Yes, we use the same definition of the neighborhood.

P6L3: "between": which normals are compared? Is it the nearest normal in the i and j grain? This would help to show diagrammatically

This is explained in the following sentences of this paragraph.

P6L5: "equals" should be "equal"

Done

P6L6: "60 degrees throughout the paper". But why 60? And this is a free parameter so I guess not "throughout the paper". But I don't understand "in the following" here. "Following" where?

Corrected. This is indeed a free parameter. We have removed "in the following" which is not required.

P6L6: Curved how? I'm having a lot of trouble understanding this criterion. I think all of these criteria need a schematic figure.

See our answer to the main comment 1.

P6L7: Delete "Therefore"

Done

P6L19: the objective here is to merge small and large grains?

Not necessarily.

P6L20: "This number of …" don't understand. greater than 10? greater than or equal to k? what is k? is k=20?

As mentioned in the sentence k is the number of nearest neighbors. A second condition is than nmin must be greater than 10, the number of parameters to fit an ellipsoid. We have rephrased the sentence to clarify it.

P6L24: I guess this is another step that needs very close operator control and field knowledge. I see the algorithm works quickly, but I imagine that this requires a lot of going back and tweaking parameters until a "satisfying" (qualitative) segmentation is reached. This probably takes a while even on a small patch with <1,000 grains as here, and would be difficult to upscale to large areas or to finer point cloud spacing (i.e., SfM at hundreds to thousands of points per cm2) where you would be able to get smaller grains and thus exponentially more segmentation would be performed (log-normal GSD typical of river sediments). SfM is used later on, but this point about the extremely high density of SfM point clouds and the exponential increase in smaller grains is never made? Or I may have missed it.

Yes, G3Point is based on a trial-and-error approach to constrain the parameters, as it is already mentioned in the manuscript.

As discussed in section 4.1, G3Point is designed to work efficiently on patch-scale studies (i.e., surface area 1-100 m2 with a typical point density of ~0.1-1 cm/point and a total number of points around $10^6$). As in most model, there is a tradeoff between resolution and size due to a finite amount of computer resources. Increasing point density, and therefore the ability to detect small grains, requires decreasing the scale of the "patch". Conversely, decreasing point density (and increasing the patch scale) allows the user to focus only on coarse grains. We have better explained this in section 4.1.

P6L27-28: "respectively" is maybe missing, if you are referring to the beginning of the sentence.

Done

P6L32: Why wasn't the example in Figure 2 cleaned? It wasn't required because the segmentation worked so well?

Yes, the segmentation was deemed suitable. This also demonstrates that the cleaning operations of the segmentation are not systematically required.

Figure 2 caption: "fit" not "fitted", here and elsewhere in the main text. "same color as in panel c". "relative" not "relatively"

Done

P8L6: "consists of"

Done

P8L7: Don't understand the use of "adequacy"

It was removed for clarity.

P8L9-10: "From directly using the point cloud to describe each grain and measure their sizes and orientations."

Done

P8L10: Delete "the use of"

Done

P8L22: Delete "Anyhow", too informal

Done

P8L28: It's not really possible to see anything in Figure 2. You may need another figure with a zoom-in on a couple of grains (the original points, colored by grain label) and the ellipse fit to the points. This may be hard to visualize, but as it stands you cannot see what is being referred to here in Figure 2. cf. Appendix Figure A1. This is going in the direction of a useful figure in panel b, but the points should be increased in size and the model fits made semi-transparent. I realize these figures are screenshots from cloud compare, but I wonder if they could still be somewhat improved by zooming in more and increasing the point size / adjusting the colors?

We have made numerous tests to provide a figure with a suitable view, which remains a challenge when representing 3D data on a 2D figure. The density of points is so high that a readable figure would focus only on a few tens of points. This would not allow to illustrate the effects of the numerical step sequence on the grains segmentation t and modeling.

P8L29: "best solution". What specifically was tested besides the least squares?

We do not think that going into the details of the pros and cons of the different ellipsoidal fitting methods is particularly relevant for this paper. Yet, to answer this comment, we have also the nonlinear Koopmans method and a direct-iterative least-square approach.

P8L30: Could you provide a reliable citation for "inertia ellipsoids"? I was doing some internet searching and having trouble understanding what these are and how they are computed. I don't get the relation between the equation of an ellipsoid and this model / mathematical operation. Even an internet link would help me here! I just don't get it, but maybe I need to accept it as mathematical fact.

We are using the matGeom/Geom3d Matlab library developed by David Legland (INRIA). The inertia ellipsoid (sometimes also called the equivalent ellipsoid) is defined here: https://mattools.github.io/matGeom/api/matGeom/geom3d/equivalentEllipsoid.html

We were not able to provide a classical citation to a paper defining the concept of inertia ellipsoid.

Some explanations on the link between SVD of random points and the fit of an ellipse in 2D are for instance given here: https://towardsdatascience.com/understanding-singular-value-decomposition-and-its-application-in-data-science-388a54be95d

P8L33: I think singular value decomposition was already defined. Fine to use "SVD"

Done

P9L5: how were the "ellipsoids" and "cuboids" compared? you mean the dimensions of the cuboid (length / width / height) with the axes of the ellipsoids (a /b /c)? Please be explicit here.

Yes. We have added a sentence to clarify this.

P10L3: "axes"

Done

P10L5: "approximates"

Done

P10L11: "metric"

Done

P10L20: "diameters"

Done

P10L27: I like the lab experiment, with the pebbles from the hardware store, but I wouldn't call this "synthetic". Rather "lab experiment" is more appropriate. Or you say "Lab Environment" and "Natural Environment". Please remove the term synthetic from the manuscript. Synthetic implies artificially generated, but these are real pebbles you are laying out and measuring (in a lab setting).

Corrected

Figure 4: The cuboid volume is definitely way far off from the true volume, but putting it on the x-axis of panel g here implied this is validation data. If you really wanted to get the volume you could do a convex hull of the labelled point cloud and directly measure the volume from this as a control to compare the ellipsoid volume with. Actually, it looks like you do exactly this in Appendix A! Another reason to put Appendix A at the beginning of the results.

If we agree that the cuboid volume is not the proper volume of the grain, we prefer to keep the cuboid volume on the x-axis to remain consistent with all the other metrics (e.g., length, axis ratio) that are displayed on this figure. We disagree that taking the volume obtained from the convex hull of the labelled points would provide a better estimate. Indeed, the grain surface is only partly covered by the point cloud and the "hidden" face of the grains is probably significant. The volume of the convex hull would necessarily lead to a underestimate of the true volume. Note that in Appendix A, the volume of the convex hull is considered as each grain was fully captured in lab conditions using photogrammetry

Figure 4 caption: "top", "middle", "bottom" should be "left, center, right".

Done

P11L11: This experiment does not test the watershed segmentation at all since the pebbles are not overlapping. Or at least, this is barely a test. That's important. The first experiment is really ONLY for testing the ellipsoid models and axis length measurements as far as I can tell. Be explicit about this point.

This first experiment is about detecting grains in a lab experiment and to properly measure them. This is what is described in our manuscript. We agree that is not a challenging test for G3Point.

P11L12: "captured by pictures" should be "photographed"

Done

P11L13: "Processed" how? Be specific. High resolution? How many interest points? Was any filtering done? What was the resolution of the input images and what camera model was used to take the photos? How many oblique vs. nadir photos and approximately how low of an oblique angle was used?

We now provide some information on the pictures used for this point cloud. We do not wish to describe in details the parameters of Agisoft Metashape as we consider this would blur the message of this (already long) paper.

P11L14: How as the planar surface removed? Manual cropping in CC? If this surface is removed, then the watershed segmentation is really barely necessary, so again this isn't testing the segmentation algorithm -- just the model fits.

The planar surface was removed by removing points below a threshold along the vertical coordinate. This is now explained.

P12L10: I wouldn't call these consistent. Maybe it helps to include a metric like RMSE of the two methods compared to the cuboid and plot this number (or two numbers) on each of the subplots in Figure 4e. It actually looks like the bias increases at larger grain sizes, why is that?

We do not wish to add more complexity to this figure that is already quite rich in information. Plotting one metric against another one is a classical way of showing potential mismatch in between these two metrics.

P12L15: The discussion of the DLSF and IE models is really interesting. I wonder if this is better highlighted in the Appendix A section, which I recommend becoming a new section at the beginning of the results.

We agree with this comment and that Appendix A should appear in the Results section (see response to comment 7).

P12L20: I'm not sure about using cuboid volume in any comparisons, this is not a volume of the grain (actually it's really far from the volume of a grain contained in the bounding box).

We agree with this comment. Yet, we also note that we simply show the cuboid volume and do not present it as a reference value nor do we make in-depth comparisons.

Figure 5: Struggling with these CDFs. Maybe make the G3P lines dashed in the plot and legend?

Done

Figure 5 caption: "envelopes"

Done

P14L1: So this is equivalent to the cuboids used in the first experiment? May be interesting to note whether the cuboid dimensions did in fact correspond well with manually measured dimensions of the hardware-store bought pebbles. No need for a formal analysis, but was this checked ever? Do the cuboids accurately give the hand-measured lengths?

We do not understand this comment as this sentence describes the Wolman grid we defined to sample the grains at Site 1. The cuboid model was applied only for the lab experiment, in order to recover in a simple way the three diameters of the grains.

P14L2: "wo"?

Corrected ("two")

P14L5: Why no a and c axes here? Time constraint? Just curious.

Indeed, it was a very warm day so for the sake of time and operator's health, we measured only one diameter on that site. We have added a sentence in that sense "The others diameters were not measured due to time constraints."

P14L9: This is a fair approach, but it would also be nice here to cite the binomial modeling approach of Eaton et al. (2019) and its application in Purinton and Bookhagen (2021).

The bootstrap approach is one common method to estimate uncertainties. We agree that other choices, such as a binomial approach mentioned by the reviewer, would perform well to. However, we do not wish here to review other possible methods related to Wolman counts so we refer to these papers in the Introduction only.

Table 1: Thanks for noting the final parameters, but what were the steps in their selection (what other values were tried?). Is this just something the operator needs to "get a feel" for? Or are there good reasons for these values? This goes towards one of my major comments regarding the parameters.

The values must be chosen according to the point cloud resolution with respect to the size of the grains, and according to the geometry of the grains themselves. The default values of G3Point should do a good initial work for any point cloud with a sub-millimetric resolution describing centimeter grains with angular to rounded shapes (such as the ones we use). If not, the good way to find the best values is to look at the first order characteristics of the point cloud: how many points for a grain? How many points between two grains? The new section in the supplement material together with appendix A should make this clearer.

P14L19: What is the MP resolution of the images?

Added (24 Ma pour Chateau Renard and 11.6 Ma pour The Hérault)

P14L19: Here and elsewhere in the manuscript it should be "square meters" not "squared meters".

Corrected.

P14L23: "trial-and-error" here and elsewhere is vague. Can you be more specific about how these tests were run? How were initial values selected and how were the modified based on the runs? Was there systematic adjustment of the parameters or was it a bit more "random"?

See previous comment on the parameter choice.

P15L1: "grains are detected"

Done

P15L5: Again, incorrect use of "synthetic". You are measuring real grains on the point cloud.

Changed for "virtual"

P15L13: "percent"

Done

Figure 6: Why only considering these percentiles? Maybe It would be cool to have a QQ plot where you show several percentiles plotted against each other for the Wol vs. G3P, with one QQ plot for each site and for each axis (so a 3x3 figure with 9 QQ plots, well 7 plots actually because Herault is missing a and c). Then you could plot e.g., the 5, 16, 25, 50, 75, 84, 95 percentiles against each other in each plot (common grain size percentiles). Could also include the common bias and accuracy metrics on the plot (cf. Purinton and Bookhagen 2019, Figure 12). For each percentile the uncertainty could also be visualized with vertical and horizontal error bars. Would be useful and you could drop Table 2.

We modify the figure following this suggestion to present more quantiles, as it eases the comparison between the two distributions. Uncertainties are given as errorbars.

P15L22: "quartile" should be "quantile" here and elsewhere. Quartile refers to the 25th, 50th, 75th, and 100th percentiles. Alternative to using quantile, consider just using the term "percentile" which is more common in grain-size studies. Quantile is usually the 0.1, 0.2, 0.3, etc., whereas percentile is 10, 20, 30.

Done

Table 2: I suggest to remove this. A Figure with vertical and horizontal error bars as I suggest in place of Figure 6 would be much easier to read and present the results more strongly.

We moved Table 2 to Supplement as we believe it is of interest to have access to values.

P17L5: Did you mean "point/cm2"? is this supposed to be a point density? or is this a distance between points?

Corrected

P17L6: "10^6". But the SfM datasets had way more points right? On the order of 10^7-9 is what I would expect from SfM point clouds of pebbles. I saw you reduced the number to "speed processing", how long would the processing have been at full resolution? Days?

Indeed, we mention in the text that we subsampled the point clouds for speed considerations, but also because the native resolution is higher than needed. We performed a few tests at full resolution, and it was in the order of a dozen of minutes. This is still very manageable but a bit annoying.

P17L19-20: "Suitable values…" A bit more guidance should be provided. what would be interesting is to show (maybe in the supplement) a zoomed in area of the point cloud with labels resulting from e.g., k = 10, 20, 30, 40, 50, to see how this effects the segmentation.

See our answer to Comment 1.

Section 4.2: By reducing the density of SfM points you are removing one of the key benefits of this method: denser point clouds! That should be noted. Point density could be very useful for measuring smaller grains...

Point clouds are still quite dense! For the applications developed here (size, sphericity, orientation), the resolution we used is largely sufficient. We believe we were a bit vague on the question of resolution in the previous version of the manuscript. In the revised version, we make it clearer that the resolution of the point cloud should be chosen with respect to the size of the grains of interest. If there is a large contrast in grain sizes at the study site, the best solution may be to split the point cloud into coarse and fine patches and to adjust the parameters to each sub point cloud.

P17L24-26: Yes! we have noted this too on SfM point clouds. This may be a Metashape issue, or an issue with the number of photos and the angle of the photos. It should be noted that the source of these issues may be related to the quality of photos taken (lighting, resolution, blurriness) and/or lack of sufficient coverage / view angles -- it is not necessarily a result of the underlying SfM algorithms, but we need to explore this more.

We agree. We have added some details on this point.

P17L29: "built-in"

Done

P18L4: Be consistent of your use of commas with "e.g." and "i.e.", in other places you do not use a comma after.

Done

P18L12: "> 10^2"... well that depends on the size of the grains versus size of surveyed areas and the density of the point cloud. It's tricky to put a number on this.

This sentence is a comparison to Wolman counts and we implicitly made the hypothesis that the sample was taken over the same area. For clarity, this is now explicit.

P18L13: Here and elsewhere I think you mean "representation", not "representativity" (not a word).

The correct word seems to be "representativeness". It has been corrected here and elsewhere.

P18L14: Yes! Here you could mention that an entire surface of the grain is retrieved, so we are not limited to fitting ellipsoid models (though this is still useful w.r.t. historical approaches).

Done

P18L16: Well, drone usage in this case is really challenging, and maybe not realistic, see Section 5.4 in Purinton and Bookhagen 2019.

We agree that drone usage can be challenging and leads to lower resolution than SfM or Lidar data. However, a cm-scale resolution can be satisfying for coarse environments such as a rock fall. Therefore, we did not modify the text.

P18L18: "hundred grains"

done

P18L21: "spent in"

done

Section 4.5: Good that you make some of these statements, but I think they can be teased earlier. You are not just limited to the a/b/c axes when we start labeling point clouds, but a wealth of other more accurate information about the grain shape (cite Domokos' work here).

We believe this (teasing about other metrics, in particular shape and orientation) is already what we do in the title, introduction and when we describe the methods and the results.

P19L22: Note, orientation is also provided by 2D image segmentation methods (e.g., PebbleCounts), although this is only the "azimuth" and not the "dip".

Corrected

P20L26: I think "select" not "generate". Why only 1000 grains? Why not use all grains?

In the previous version of the manuscript, the algorithm was run once for the DLSF method and then a second time for the IE method. The average distribution was then calculated from these two distributions. As a result, the average distribution was not attached to any real grain. This is why we needed to generate grains from the average distribution but this was a bit of a pity considering that we do have access to the grains. Therefore, we have updated the algorithm so that each grain is fitted by the two approaches in one run. We can now calculate the average model for each grain instead of doing it on the distribution. By doing so, the weak difference in sphericity between the 3 study sites appears even weaker. Therefore, we remove this part from the revised version of the manuscript.

Figure 9 caption (and Figure S7 caption): Check your definition of the boxplot. If 100% of the data was always in the whiskers there would be no red outliers outside the whiskers.

This figure has been removed.

P21L13: "solves the issue" is a little strong here. I would say "makes progress on the issue".

done

P21L14: I think "methodological advance" is also a little strong. Rather, the authors take the watershed approach concept used by many other algorithms / studies and apply it to point clouds in the framework of a network-based approach. This is an "alternative" and a "unique application to point clouds", but I would refrain from "advance" w.r.t. other studies that use different approaches.

Changed.

P21L19: Not sure what is meant by this last statement, maybe rephrase.

Done

P21L20: Yes, it detected all grains in the lab experiment, but is that an interesting or notable result? The grains were far from each other and not overlapping (and the area between them was removed manually I think?). The lab experiment was more about testing the ellipse models.

Corrected.

P22L7-8: Shouldn't this statement be in the discussion?

We see it more as an opening statement than a discussion.

P22L9-13: This is really late to be introducing previous and highly relevant work. A discussion of this alternative approach should be up in the introduction or methods section. I agree with the point about G3P used to generate training data, but that could go in the discussion.

We now mention Chen et al. (2020) in the introduction and discuss it in the Discussion.

Appendix:

P29L1: This appendix is enormously important! I think it should really be a section at the beginning of the results. It really helps my understanding of the models and the limitations based on grain shape. Grain shape is extremely important if you also make the statement about the ellipsoid model being an imperfect descriptor of natural fluvial sediments

We agree. See response to comment 7.

P29L7: "consist of"

Done

P29L9: Be specific, how were all sides of the grains collected? Was the object rotated to get a photo from every side? were they laid on a flat surface (so there is still part of the grain "missing")?

Done

P29L10: I'm not sure if you just want a percentage or a rather what is more interesting is taking a percentage from a certain region, or dropping an area of points to simulate an occlusion. Is that what is done? Not clear from this description.

This is exactly what we do and what we explain in the following sentence.

P29L12: "seeked"?

Corrected.

P29L15: Convex hull is used here! It should also be used in Figure 4 rather than "cuboid volume".

We disagree. Here, we have access to the entire grain surface, which is not the case in Figure 4 or elsewhere. This is why we can use the convex hull in this case and not in the previous ones (where we would miss a significant – but hard estimate – volume).

Figure A1: Figure is pretty low-resolution. It would be nice if when I zoom in on the panel b I can see the points and ellipsoids, but the resolution is too low for that right now. Not strictly necessary, but this would be really helpful.

The figure we have provided has a much better resolution than the one available on the pdf, and allows the reader to see the points and the ellipsoid. This is unfortunately not something we can change by ourselves.

P30L14: But these ellipsoidal models are likely increasingly "wrong" for more angular grains. I don't think you need to modify the analysis but this point should definitely be noted! I think the path forward for grain-size measurement from point clouds does not lie in ellipsoidal models, but rather measurements directly on the labelled points.

The role of this appendix is to test the ability of the different ellipsoidal models to represent the size and geometry of the grains.

P30L15-16: This is important! DLSF overestimates c. We see that in Figure 4. That's why this appendix / figure should come at the beginning of the results.

See response to comment 7.

P30L17: "assuming that"

References

Purinton, B., & Bookhagen, B. (2021). Tracking downstream variability in large grain-size distributions in the south-central Andes. Journal of Geophysical Research: Earth Surface, 126, e2021JF006260. https://doi.org/10.1029/2021JF006260

Rheinwalt, A., Goswami, B., & Bookhagen, B. (2019). A network-based flow accumulation algorithm for point clouds: Facet-Flow Networks (FFNs). Journal of Geophysical Research: Earth Surface, 124. 2013–2033 https://doi.org/10.1029/2018JF004827

Smith, M., Carrivick, J., and Quincey, D.: Structure from motion photogrammetry in physical geography, Prog. Phys. Geog., 40, 247–275, https://doi.org/10.1177/0309133315615805, 2015

**Line or specific Comments (Anonymous Reviewer #2)**

The reviewer also has the detailed comments, concerns, and suggestions as follows.

Page 1 line 5-10: Please rewrite the sentence: "The grain-scale morphology of sediments and their size distribution inform on their transport history, are important factors controlling the efficiency of erosion and transport and control the quality of aquatic ecosystems."

Done

Page 1 line 5-10: This is a redundant sentence: "In turn, constraining the spatial evolution of the size and shape of grains can offer deep insights on the dynamics of erosion and sediment transport 10 in coastal, hillslope and fluvial environments."

Done

Page 1 line 20: in-situ is a misused word.

We are happy with the use of this latin expression.

Page 1 line 25: it should be polygonal instead of polyhedral

We disagree. A polygon is a 2D geometric figure, while a polyhedron is a 3D geometric figure, which is what we intend to mean.

Page 2 line 0-5: this sentence is confusing. Please clarify or rewrite it. "The size and shape distribution of grains in various natural environments can therefore be represented as an initial size or shape distribution, informing on fragmentation, weathering processes and on the structure of the rock mass (e.g., fracture density and orientation, mineral size)"

Modified

Page 2 line 20: untrue statement: "the 3D geometry of grains and their statistical distributions in natural environments remain poorly known." Either 3D geometry of grains or grain distributions have been researched. The authors should do more research on this.

We fully agree. We have changed this sentence to say that documenting the 3D geometry of grains and their distributions remains a challenge.

Page 2 line 25-30: recent grain segmentation based on machine learning and UAS has been applied to a large scale (hundreds of meters).

Yes, that is what we already mention in the following paragraph.

Page 3. Please include more literature reviews as mentioned above.

See our response to comment 9.

Page 3 line 15-20: segmenting grains from airborne lidar is doubtful. Airborne lidar only has resolution of meters, which is much worse than SfM.

We partly disagree. Airborne LiDAR can offer some constraints to help detecting and measuring the size of large boulders (>1 m) driven by large landslide or floods. Numerous recent papers in geomorphology focus on the topic of the role of this boulder in controlling river erosion and sediment transport.

Page 3 line 25: actually what this study has obtained is histograms instead of distributions.

We disagree with this statement. G3point outputs distributions (Figure 5) that can then be shown in various ways, such as histograms (Figure 3) or whisker plots (Figure 9).

Page 4 line 0-5: it's good the authors mention the assumptions. However, another assumption is that there are no vertically stacked rocks.

We have added a sentence to mention this limitation.

Page 4 line 15-20: what is the context of "local minimum"? Because local minimum is used in optimization problems, the authors should clarify its usage or metrics here.

We now mention "topographic minimum".

Page 4 line 20: missing introduction to the Fastscape algorithm. E.g. How does it work? In the same sentence, "order" is an ambiguous word.

We now briefly describe the Fastscape algorithm for node ordering and watershed identification.

Page 5 line 1: as long as the neighborhood nodes of each node are known. Plural

corrected

Page 5 line 1: the 3D distance used in the method is unspecified. E.g. Is it Euclidean distance, Manhattan distance, or anything else?

This is Euclidean distances, as now stated in the manuscript.

Page 5 line 5-10: support or evidence is lacking to state "each grain is theoretically identified by a single watershed"

We have modified the sentence and now state that "each grain should be identified".

Page line 10: please specify the laptop configuration used in the study. E.g. processors, RAM, etc.

Done (laptop with 32 GB of RAM and a Intel i9 CPU of 8 cores with a clock speed of 2.4 GHz)

Page 6 line 1: the 3D angle is unspecified. Is it angle in degrees or radians, Euler angle, dot product, quaternion, or anything else?

The definition of a 3D angle is standard (angle between two 3D vectors along the plane defined by these two vectors). We later specify it is measured in degrees.

Figure 2: the review suggests to include a detailed, zoom-in figure to demonstrate the merging results.

See response to comment 1.

Page 6 line 15: not scientific writing: "If this initial segmentation is deemed satisfactory at first order, some minor flaws can lead to an inaccurate description of the geometry of grains and their size distributions."

The sentence has been removed.

Page 6 line 20: point cleaning process is conducted to remove segments with a few points. However, the abstract states the method is only limited to segmenting grains greater than one 3D point. These two are contradictory.

We do not understand this comment as we do not mention in the abstract that "the method is only limited to segmenting grains greater than one 3D point" (or similar).

Page 6 line 25-30: please provide intuitive explanation of "a minimum or an intermediate singular value divided by its maximum singular value"

Done.

Page 8 line 25: If the constraint has not been considered in the fitting, then the fitting should be called quadrics fitting instead of ellipsoid fitting.

The method we use is intended to fit ellipsoids, so we prefer to maintain this description.

Page 8 line 25-30: support or evidence is lacking to state "Other ellipsoidal fitting algorithms exist, but this direct least-square approach was found to lead to the best solution."

We meant "for the data we used". This has been corrected in the revised version.

Page 10 3 Result. The paper states the method can be applied to various point cloud sources including lidar. However, they haven't conducted any experiments on lidar. For example, airborne lidar has lower resolution. It is doubtful that their method can be applied to airborne lidar.

We disagree with this statement as the Otira point cloud, used to present the algorithm, has been acquired with terrestrial Lidar (section 2). The method can be applied to any point cloud providing that the resolution is adequate for the sampled grains: in section 2.3, we indicate that a grain should be described by at least 10 points (this is considered as the minimum number of points required to fit an ellipsoid. Yet, larger numbers of points should be favoured to reduce the uncertainty on the fit. We add in section 4.3 of the revised version "The point density should be high enough so that each grain is described by at least several dozen of points.".

Page 12 line 5: It is unclear why the cuboids are used as ground truth.

To clarify, we add "As the grains are lying flat, the length and the width of the cuboids correspond to the long and intermediate axes of the grains, respectively."

Page 13 Figure 5 b) and c): the study sites for Wolman method and SfM are not the same. If the data sources are different, how can the authors guarantee it is meaningful to compare the grain segmentation results? Please clarify this point or consider the same area for the experiment.

We selected the sampling sites to avoid grain size trends at the surface so that we can directly compare the two approaches. The samples were taken at the same location, but their extents are indeed not exactly the same (the SfM extent covers about half of the Wolman count extent, as a reduced SfM area insures a lighter and thus more manageable data set). However, due to the homogeneity of the study sites, this not a first order issue. To illustrate this, we cut our Wolman samples and select only the grains that were closed to the SfM area. We compare these partial samples to the total samples on the figure below. There is no significant impact on the distributions themselves except for the lower quantiles on Site 2. The main impact of reducing the Wolman samples is to increase the uncertainties because there are less grains. Therefore, to keep the highest possible precision, we keep the samples as they were.

[Figure]

Page 15 line 5: please explain more details about the Wolman sampling method. Is it a manual or automated method? Is it 2D or 3D?

We now specify "we apply a virtual grid on the 3D point cloud and automatically extract ".

Page 15 Figure 6: why are the results from Herault missing?

There is only one diameter for the Herault (b-axis), plotted on the right on the previous version of Figure 6. We hope the new figure is clearer.

Page 17 line 5-10: another important reason to use "patch-scale" areas is that the proposed method is not effective for high-frequency terrains.

We do not understand the meaning of this comment and in particular of the expression "high-frequency terrains"

Page 17 line 5-10: It is doubtful that G3Point can be applied to large areas greater than 100 meter squares. There are many reasons. For example, the method is limited by the computer's RAM capacity. It is unclear how efficient the watershed algorithm is to segment 3D points. Its time complexity is unknown. Also, searching neighborhoods is also time consuming.

From an algorithmic point of view, there is no restriction in terms of sample area. Time might indeed be a limitation for very large samples with high point density, yet, the segmentation is fast (see Section 2) so that the grain-size distribution can be obtained in a short amount of time (see section 4.3). Therefore, we do not consider computer capacity and time as a limit to G3Point. However, we recommend to visually check the quality of the segmentation and this can be challenging on large point clouds. In addition, grain size may vary with space. These two considerations lead us to recommend to split large samples in order to work at the "patch-scale".

Page 17 20-25: untrue statement: "Point clouds obtained with LiDAR data provide better accuracy than SFM but can be associated to varying resolution, while the ones obtained by SFM provide uniform resolution but can lead to some inaccuracies." Only terrestrial lidar can have higher accuracy than SfM. UAS lidar or airborne lidar have worse local accuracy than SfM

To clarify this point, we now mention "Point clouds obtained with terrestrial LiDAR …"

Page 18 4.3. The comparison between previous 2D grain segmentation and the G3Point 3D grain segmentation is missing.

As stated in the introduction, 2D methods are inherently bias by the 2D exposure of 3D objects. This may induce bias in the grain size distribution and such methods can't access the 3D geometry of the grains (sphericity, azimut, etc). These limitations have been addressed by previous workers (Bunte and Abt, 2001 and references therein, Graham et al., 2010). We choose to refer to their work rather than showing additional analysis for the sake of time and manuscript's length.

References

Thanks for these references, we added them to the revised version of the manuscript.

Butler, J. B., Lane, S. N., & Chandler, J. H. (2001). Automated extraction of grain-size data from gravel surfaces using digital image processing. Journal of Hydraulic Research, 39(5), 519–529. https://doi.org/10.1080/00221686.2001.9628276

Carbonneau, P. e., Bizzi, S., & Marchetti, G. (2018). Robotic photosieving from low-cost multirotor sUAS: A proof-of-concept. Earth Surface Processes and Landforms, 43(5), 1160–1166. https://doi.org/10.1002/esp.4298

Langhammer, J., Lendzioch, T., Miřijovský, J., & Hartvich, F. (2017). UAV-Based Optical Granulometry as Tool for Detecting Changes in Structure of Flood Depositions. Remote Sensing, 9(3), 240. https://doi.org/10.3390/rs9030240

---

## Author Response (AR2)

Comments by Associate Editor (Rebecca Hodge)
Many thanks for undertaking these edits on the paper. Both the original reviewers have provided a set of comments on the revised paper and agree that the edits have improved the paper. The new comments are mainly points of clarification and should be straightforward for you to address. Referee #2 has provided an extensive list of copy edits. You should consider all the suggestions, but it's up to you to decide which ones to implement. Personally, I don't think that all of them are necessary. I do not agree that the paper requires significant improvement in scientific writing, and note that the paper will undergo copy editing prior to publication anyway.

We thank the Associate Editor for this positive evaluation of our work and these recommendations. We have in turn modified our manuscripts following these guidelines.

Comments by Referee #1 (Benjamin Purinton)

Dear Authors,

I applaud the effort revising the paper. In particular I find the qualifications of the trial-and-error approach and additional schematic figures (Fig. A1 is great, thank you) and text to be very helpful in understanding the steps and limitations of the algorithm, and this will certainly aid other researchers when they are seeking to apply this. I briefly tried the matlab code and it seemed to run without errors (though admittedly I did not closely inspect the resulting segmentation). The paper is ready for publication following some minor changes/additions that I detail below.

Sincerely,
Benjamin Purinton

We are very grateful to Benjamin Purinton for his work on our manuscript which has clearly helped increasing its quality. We have modified our manuscript to account for these minor comments.

Supplement:

The phrase is "on the contrary" (noted a number of times). Maybe rephrase "dichomotic approach", I don't quite understand what you mean. Typo: "Figure S38b".
Corrected

Main Text:

There is a recent citation that would be good to include in Section 4.3 in a couple of sentences. I think it's vital that this paper is cited as it presents an alternative 3D segmentation approach to G3Point:
A. Walicka and N. Pfeifer, "Automatic Segmentation of Individual Grains From a Terrestrial Laser Scanning Point Cloud of a Mountain River Bed," in IEEE Journal of Selected Topics in Applied Earth Observations and Remote Sensing, vol. 15, pp. 1389-1410, 2022, doi: 10.1109/JSTARS.2022.3141892.
Thanks for pointing out this ref, that we now mention in the introduction and in section 4.3.

P6L25: should be "give water"
Done

P8L6: "curved" how? Maybe it would be better to write "concave"? Although that would mean alpha should be > 45 degrees? In which case maybe just note that > 45 degrees indicates a concave

border.. If I'm understanding this correctly. Basically, a little guidance / explanation about concavity vs. convexity regarding this angle and the merging would be helpful.

Actually, we do not compute the curvature, but the change in normal orientation (even if the two are obviously related when relating our measure to a distance). We have slightly rephrased this sentence to try clarifying it.

P8L18: Same as above, some guidance on how beta relates to convexity / concavity of the borders.
It is the same measure than the one performed for Criterion 3.

P8L21-22: "greater than or equal to", two locations it is worded incorrectly
Done

P9L3: "Figure" capitalized
Done

P14L29: Missing a "x" multiplication symbol in "~2 10^5"
Done

P16L7: "constrain" should be "constraint"
Done

P19L16: "a q-q diagram" or "q-q diagrams"
Done

P20L27: "study areas"
Done

P22L8: "time spent"
Done

P22L11: "provide, in a few minutes, thousands"
Done

P24L22: "If both models accurately infer of the major and intermediate axes"... rephrase, I don't understand this.
Done

Comments by Referee #2 (Anonymous)
The resubmission addressed most of the previous comments. The manuscript structure is clear, and the writing is much better. However, I still have two major comments.
We are very grateful to the second referee for his work on our manuscript. We have modified our manuscript to account for some the comments identified.

It is unclear how G3Point conducts grain segmentation. First, how does G3Point select the initial points to start grain segmentation? I.e., do you select every point in a point cloud to find a path to the summit? Or do you randomly sample points to find their summits? If I understand correctly, based on the description of the Fastscape algorithm, only a path or a stream is found. The adjacent nodes in the path have the steepest gradient. How does the path with a list of nodes result in a segmented patch? It is important to clarify this point because the computing efficiency is related to the number of summits, which are the results of the grain segmentation.
The Methods section already addresses these comments. In anutshell: the Fastscape algorithm is applied to all the points of the point clouds (there is no random selection of some points). Fastscape

generates one acyclic graph for each watershed, which relates each point of this watershed to the summit (the "outlet" node). Each watershed is therefore readily obtained and provide a labelisation of each grain.

The quality of this research is good for an ESurf publication. However, that's only after significant improvement in scientific writing. I appreciate the proposed methodology, validation, and discussion. Although the writing of this resubmission has been improved, publishing this work still requires much more work. Please see the minor comments for some suggestions.
We below account for most of the minor comments identified by Referee #2, as suggested by the Associate Editor.

Minor comments.
Title: 3D point clouds: 3D is redundant because it is implied in point cloud.
We agree that it is a bit redundant but using "3D" and "point cloud" makes it very clear that the point cloud must be a real (x,y,z) and not a 2D raster. Therefore, we kept "3D point cloud".

P1 L7: The grain-scale morphology and size distribution of sediments are important factors controlling the efficiency of erosion, sediment transport and the quality of aquatic ecosystems.
>> The grain-scale morphology and size distribution of sediments are important factors controlling the erosion efficiency, sediment transport, and aquatic ecosystem quality.
Done

P1 L8: In turn, constraining the spatial evolution …. what do you mean by "constraining"? It does not make sense. Do you mean "obtaining"?
Changed for "characterising"

P1 L12: methodological approach >> methodology
Done

P1 L13: geomorphological >> geometric
Done

P1 L14: remove "applied here to 3D point clouds". Redundant
P1 L15: remove "applied to each sub-cloud". Redundant.
We kept these two sentences to prevent any confusion.

P1 L16: "conceived" is an inappropriate word here
Removed

P1 L23: I understand why the authors want to use this word, in-situ, but in-situ has a specific meaning when talking about sampling and measuring method. In this case, it is better to avoid such ambiguity. Please refer to google scholar results of in-situ:
https://scholar.google.com/scholar?hl=en&as_sdt=0%2C3&q=in-situ&btnG=
We believe there is no possible confusion in the context of the paper and therefore, we did not change the text.

P1 L23: remove "and grain cluster". Redundant with grains
Done

P1 L24: "The main limit of this method is that it is only able to detect grains with a characteristic size significantly greater than the resolution of the point cloud". This limitation is very obvious.

Authors should be more open to unique problems of the presented method. Or you can remove this sentence.
We had feedbacks from the first reviews and from people using G3Point that this limitation was not that obvious for users. We therefore decided to make it very explicite and we kept this sentence in the revised manuscript.

P2 L1: remove "on the"
Done

P2 L20: potentiality >> potential
Done

P2 L21: documenting >> inventorying
We believe documenting is the correct word here

P2 L23: remove "most". You need to conduct comparison or include reference to make "the most" statement.
Done

P2 L23: "This method consists in measuring" >> This method measures. "consist in" is overly misused in the entire manuscript. Please modify them accordingly.
Done

P2 L24: Suggested writing: The grid-by-number method is simple to implement and similar to a volumetric sampling xxx
We removed the commas and "which" to simplify the sentence.

P2 L29: Collection of a data set >> Collecting a data set
Done

P3 L1: misused "consists in"
Done

P3 L24: suggested writing: Structure-from-Motion (SfM).
Done

P3 L35: Suggested writing: considering point clouds obtained from SfM to check its xxx.
Done

P4 L8: remove "any type of"
Done

P4 L9: remove "summarized by".
Done

P4 L9: remove the sentence, "A 3D point cloud …". Point cloud is not new. It is unnecessary to explain.
Done

P4 L11: remove "2". It causes confusion.
Done

P4 L13: remove "will"
Done

P4 L14: how do you deal with background points such as dirt and mud?
Sentence has been changed to clarify that the point cloud must contain only grains to be segmented.

P4 L14: this task >> this denoising task
Done

P6 L7: remove "will"
Done

P6 L14: remove "despite this main disadvantage"
Done

P6 P16: is performed using >> uses
Done

P6 P21: You may need a transition sentence before mentioning "the Fastscape algorithm". It is unclear how the Fastscape algorithm is related to the point cloud algorithm you mentioned above.
We added "To perform the watershed segmentation" at the beginning of the sentence.

P6 L21: order is ambiguous word. Do you mean "make the points in a sequence" ?
Order (or ordering) is the word that is used by Braun & Willett (2013) and by most graph librairies.

P6 L25: giver >> donor. You should have consistent terms through the entire manuscript.
Done

P7 L7: "This algorithm only imposes one scale". What do you mean by a scale? Scale is an ambiguous word.
We added "spatial".

P7 L23: Fig A1b. missing or you mean Fig S1b?
Fig. A refers to the figure in Appendix A.

P7 L25: "Instead of xxx". Long sentence. It's difficult to follow.
We split the sentence in two.

P7 L30: what k value do you choose here?
The same as in previous section, therefore we have kept the same name of variable.

P8 L8-10: If I understand this correctly, the number of summits is determined by the number of initial points. However, how initial points are selected is unspecified. This point is important because your algorithm efficiency is quadratic and thus limited by the number of summits.
We do not fully understand this comment. But no, the number of summits is not determined by the number of initial points (all the points are considered) but depends on the number of grains and their size.

P8 L17: consists in merging >> merges
Done

P9 L10: "The most pertinent and simplest xxx". how do you know this? have you done with comparison? or please cite references that draw this conclusion. Otherwise, be cautious about "the most" statement.

We removed "most pertinent".

P9 L19: respected >> satisfied

Done

P9 L30: consists in computing >> computes

Done

P11 L10: "with an error less than 1.061% when p=1.6707". how do you know? please clarify or add citation

See https://www.johndcook.com/blog/2021/03/24/surface-area-ellipsoid/ or https://www.web-formulas.com/Math_Formulas/Geometry_Surface_of_Ellipsoid.aspx

P11 L33: "lab or natural environments". Adding a data description at the beginning (after this paragraph and before 3.1), such as data quality and acquisition method, will help audience understand.

Data were acquired in different settings and with different methods. Therefore, we describe the data acquisition and quality at the beginning of each sub-section rather than at the beginning of the whole section. This prevents non-linear reading. In addition, this is a section dedicated to the validation of the method. Advices and recommendations for data acquisition are in the Discussion.

P12 L14: "For this purpose". It is unclear what purpose you are referring to.

Removed

Section 3: Your writing in describing experiment processes is much more clear than describing method ideas. The validation work is solid.

Thanks

P14 L30: "To segment grains, xxx". This is why I asked about the dirt and mud? how can your algorithm manage them? If not, you should add an assumption at the beginning.

G3Point is a segmentation algorithm, not a classifier. Therefore, the point cloud must contain only grains to be segmented. We believe there is no possible confusion in the revised version.

P20 L26: remove sentence, "G3Point can also perform grain size, xxx", unless you had experiments on such large study areas.

We did not modify this sentence because G3Point can apply on any point cloud. The algorithm can be apply on large point clouds providing that the computer has enough capacity.

P20 L30: "If G3Point can be directly applied to point clouds, xxx". I don't understand why the authors mention this assumption in the context. also, how does a user without any G3Point experience know if G3Point can be applied to their studies?

We removed the beginning of the sentence.

Section 4.1. The authors seem to talk about calibration instead of validation. I think it's better to use a known grain size to calibrate the hyperparameters as other semi-automatic approaches do. Adding a calibration process overcomes the problem of your trial-and-error exploration.

We do mean validation as there is no calibration in G3Point.

P21 L14: "removing points". how? does G3Point have any algorithms for this? or users should manually do this?

P21 L14: "which is a built-in option". What do you mean by this?

We rephrased this sentence to clarify that G3Point has an option to remove the points located at topographic minimas if needed.

P21 L20: "the point clouds processed by G3Point must be xxx". mentioning this earlier as an assumption will help audience understand.

We moved this sentence to P4.